# Negatively Correlated Ensemble Reinforcement Learning for Online Diverse Game Level Generation

**Ziqi Wang**[1,2]**, Chengpeng Hu**[1,2]**, Jialin Liu**[*1,2]**, Xin Yao**[3]
[1]Department of Computer Science and Engineering,
  Southern University of Science and Technology
[2]Research Institute of Trustworthy Autonomous Systems,
  Southern University of Science and Technology
[3]Department of Computing and Decision Sciences, Lingnan University
{wangzq2021,hucp2021}@mail.sustech.edu.cn,
liujl@sustech.edu.cn, xinyao@ln.edu.hk

## Abstract

Deep reinforcement learning has recently been successfully applied to online procedural content generation in which a policy determines promising game-level segments. However, existing methods can hardly discover diverse level patterns, while the lack of diversity makes the gameplay boring. This paper proposes an ensemble reinforcement learning approach that uses multiple negatively correlated sub-policies to generate different alternative level segments, and stochastically selects one of them following a dynamic selector policy. A novel policy regularisation technique is integrated into the approach to diversify the generated alternatives. In addition, we develop theorems to provide general methodologies for optimising policy regularisation in a Markov decision process. The proposed approach is compared with several state-of-the-art policy ensemble methods and classic methods on a well-known level generation benchmark, with two different reward functions expressing game-design goals from different perspectives. Results show that our approach boosts level diversity notably with competitive performance in terms of the reward. Furthermore, by varying the regularisation coefficient values, the trained generators form a well-spread Pareto front, allowing explicit trade-offs between diversity and rewards of generated levels.

## 1 Introduction

Continuously generating new content in a game level in real-time during game-playing, namely online level generation (OLG), is an important demand from the game industry (Amato, 2017). Recent works show that reinforcement learning (RL) is capable of training generators that can offline generate levels to satisfy customised needs using carefully designed reward functions (Khalifa et al., 2020; Huber et al., 2021). Inspired by those works, Shu et al. (2021) propose the experience-driven procedural content generation (PCG) via RL (EDRL) framework, in which an RL policy observes previously generated level segments and determines the following segment in real-time during gameplaying. EDRL is shown to be efficient and effective in generating promising levels (Shu et al., 2021; Wang et al., 2022). Then, Wang et al. (2023) show that levels generated by EDRL can be quite similar, i.e., lacking diversity. Diversity is one of the essential characteristics for levels since similar levels make players bored soon (Koster, 2013; Gravina et al., 2019). Research interests in generating diverse levels have a long history of at least two decades (Greuter et al., 2003; Togelius et al., 2011) and has been rapidly growing over the past few years (Gravina et al., 2019; Liu et al., 2021; Guzdial et al., 2022). However, to the best of our knowledge, no work has tackled the issue of limited diversity of levels online generated by RL policies yet.

---

[*]Corresponding author.

There are mainly two limitations in existing deep RL algorithms for learning to online generate diverse game levels. Firstly, existing deep RL algorithms typically use a greedy or unimodal stochastic policy. Such policy has a limited capability of representing complex and diverse decision distribution (Ren et al., 2021), thus it is hard to enable diverse level generation. Secondly, the diversity being concerned in this work is about the variations among the generated levels. In the context of OLG via RL, it is induced by the probability distribution of trajectories from the Markov decision process (MDP). A reward function only evaluates single actions, but is not aware of the entire MDP-trajectory distribution. Therefore, one can hardly formulate a reward function to express diversity.

To address the two challenges, this paper proposes an ensemble RL approach which performs a stochastic branching generation process, namely *negatively correlated ensemble RL* (NCERL). NCERL uses multiple individual actors to generate different alternative level segments. A selector policy is employed to determine the final output segment from the alternatives. Figure 1 shows a diagram of the approach. To diversify the generated alternatives, NCERL incorporates a *negative correlation regularisation* to increase the distances between the decision distributions determined by each pair of actors. As the regularisation evaluates the decision distributions rather than the action instances, traditional RL methodologies do not directly work for it. To tackle this problem, we derive the regularised versions of the policy iteration (Sutton & Barto, 2018) and policy gradient (Sutton et al., 1999) to provide fundamental methodologies for optimising policy regularisation in an MDP. Those theorems can derive general loss functions to establish regularised off-policy and on-policy deep RL algorithms, respectively. Furthermore, the reward evaluation in OLG tasks usually relies on simulating

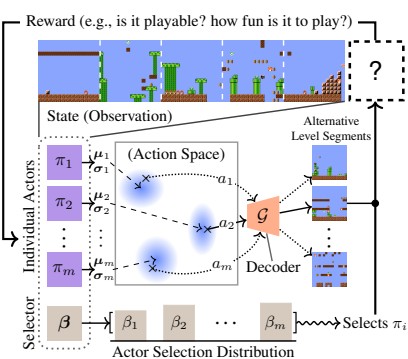

Figure 1: Overview of our approach. The decoder $\mathcal{G}$ maps a continuous action $a$ to a segment. The selector outputs a probability vector $\boldsymbol{\beta}$. One of the individual actors $\pi_i$ will be selected following $\boldsymbol{\beta}$ to generate a new segment. Section 3 details the notations.

a game-playing agent on the generated levels, which is time-consuming. Our work also reduces the time cost of training OLG agents by designing an asynchronous off-policy training framework.

Main contributions of our work are as follows: (i) we propose an ensemble RL approach with a novel negative correlation regularisation to promote the diversity of levels online generated by RL policies; (ii) regularised versions of policy iteration and policy gradient theorems are derived to illustrate how the policy regularisation can be optimised; (iii) comprehensive experiments show that by using different regularisation coefficient values, our approach produces a wide range of non-dominated policies against state-of-the-art policy ensemble RL methods, in terms of cumulative reward and diversity of generated levels. This makes it possible to make trade-offs based on specific preferences. Code and results are available at `https://github.com/PneuC/NCERL-Diverse-PCG`.

## 2 BACKGROUND AND RELATED WORK

**Procedural Content Generation via RL**   PCG has been investigated for decades (Togelius et al., 2011; Yannakakis & Togelius, 2018; Risi & Togelius, 2020; Liu et al., 2021; Guzdial et al., 2022). RL has been applied to a variety of offline PCG tasks, including level generation (Susanto & Tjandrasa, 2021) and map generation (Khalifa et al., 2020; Earle et al., 2021; Jiang et al., 2022), and shown to be powerful on those tasks. However, those works focus on offline content generation, where the content is determined before game-playing. Besides, RL has also been applied to mixed-initiative or interactive PCG (Guzdial et al., 2019), which considers the interactions between human designers and content generators. The focus of this work is generating diverse levels in an online manner. The word "online" not only refers to real-time generation but also implies the consideration of player experience while generating new content (Yannakakis & Togelius, 2011). Therefore, our work is more aligned with player-centered approaches, rather than interactive PCG or mixed-initiative systems (Guzdial et al., 2019).

Recently, research interest in online PCG via RL has been raised. Compared to traditional methods for online PCG like rule-based methods (Stammer et al., 2015; Jennings-Teats et al., 2010) and

search-based methods (Shaker et al., 2012; de Pontes et al., 2022), the RL-based methods rely on little domain knowledge and are scalable. Shu et al. (2021) introduce the EDRL framework to online generate playable, fun and historically deviated levels. Wang et al. (2022) propose a model to estimate player experience and use EDRL to train generators that optimise the experience of playing the online generated levels. RL has also been applied to specific online PCG scenarios including dynamic difficulty adjustment (Huber et al., 2021), music-driven level generation (Wang & Liu, 2022) and adaptive virtual reality exposure therapy (Mahmoudi-Nejad et al., 2021). In research of OLG via RL, the limited diversity of generated levels has been reported by Wang & Liu (2022) and Wang et al. (2023) but not addressed yet. Our work implements EDRL as the online PCG framework and specifically focuses on the RL part of it to improve diversity.

**Methods for Diverse Game Content Generation**   In offline PCG, it is applicable to formulate the contribution of an individual to the diversity and generate a set of diverse game content gradually given the formulation. A representative method is novelty search (Preuss et al., 2014; Liapis et al., 2013). Beukman et al. (2022) integrate neuroevolution and novelty search to evolve multiple generators to online generate diverse levels. Another popular method is quality-diversity (QD) search (Gravina et al., 2019; Fontaine et al., 2021), which searches for the best quality content under each possible attribute or behaviour combination. QD search has been applied to offline PCG (Gravina et al., 2019; Fontaine et al., 2021; Guzdial et al., 2022), mixed-initiative PCG (Alvarez et al., 2019) and has also been integrated with RL to train diverse policies (Tjanaka et al., 2022). However, searching for game content is typically slower than generating content via machine learning models, thus existing novelty search and QD search approaches are not likely efficient enough to realise real-time generation. Moreover, the QD method is powerful for problems with some attribute or behaviour descriptors to express diversity as the coverage over the corresponding attribute or behaviour space, while our work does not use such descriptors. Besides, Nam et al. (2021) use RL to generate role-playing game stages and use a diversity-enhancing strategy based on some rules.

**Population-based RL**   Population-based RL is used for a variety of aspects. Saphal et al. (2021) select a diverse subset from a set of trained policies to make decisions. ACE (Zhang & Yao, 2019) combines tree search and policy ensemble for better sample efficiency and value prediction. A number of works consider the population diversity in policy ensemble to encourage the exploration. SUNRISE (Lee et al., 2021) integrates several enhancement techniques into ensemble RL. PMOE (Ren et al., 2021) leverages multimodal ensemble policy to improve the exploration ability of RL agents. Parker-Holder et al. (2020) define population diversity as the determinant of an embedding matrix and show that incorporating such a diversity formulation into the training objective generally improves the reward performance of RL algorithms. Yang et al. (2022) integrate a regularisation loss to enhance the decision diversity of ensemble policy. Diversity of policy population is also concerned in multi-agent RL. For example, Cui et al. (2023) propose *adversarial diversity* to produce meaningfully diverse policies, Lupu et al. (2021) propose *trajectory diversity* to enable more robust zero-shot coordination, while Charakorn et al. (2023) propose *compatibility gap* to train a population of diverse policies. While diversity in policy populations has been explored in the aforementioned works, our work uniquely extends this consideration to the context of PCG, making diversity a primary goal alongside reward. Our ensemble policy uses a selector and multiple individual actors, which is similar to hierarchical RL (Pateria et al., 2021) and skill discovery (Konidaris & Barto, 2009). However, our method features a novel diversity regularisation.

**Regularisation in RL**   According to (Sheikh et al., 2022), regularisation methods in RL are applied for better exploration (Grau-Moya et al., 2019; Haarnoja et al., 2018a), generalisation (Farebrother et al., 2018) and other aspects that promote overall rewards (Sheikh et al., 2022; Grau-Moya et al., 2019; Cheng et al., 2019; Galashov et al.). The works of (Sheikh et al., 2022; Yang et al., 2022) study a combination of ensemble and regularisation. Different from our approach, the former regularises network parameters rather than policy behaviour, while the latter focuses on discrete action space rather than the continuous action space addressed in this work. The work by Haarnoja et al. (2018a) integrates an entropy regularisation for the decision distribution determined by the policy. Compared with their work, this work extends their theoretical results from entropy regularisation to general regularisation. Moreover, we derive the policy gradient for regularisation, which is a basis of on-policy deep RL algorithms but is not discussed in the work by Haarnoja et al. (2018a).

## 3 NEGATIVELY CORRELATED ENSEMBLE RL

This section introduces the problem formulation, then describes our proposed ensemble approach, called *negatively correlated ensemble RL* (NCERL). NCERL features a multimodal ensemble policy and a negative correlation regularisation, to address the two aforementioned limitations.

**Problem Formulation** According to (Sutton & Barto, 2018), a general MDP $\mathcal{M}$ consists of a *state space* $\mathcal{S}$, an *action space* $\mathcal{A}$ and its *dynamics* defined by a probability (density) function $p(s', r|s, a)$ over $\mathcal{S} \times \mathbb{R} \times \mathcal{S} \times \mathcal{A}$, where $s \in \mathcal{S}$ and $s' \in \mathcal{S}$ are the current and next *state*, respectively, $r \in \mathcal{R} \subset \mathbb{R}$ is a *reward*, and $a \in \mathcal{A}$ is the *action* taken at $s$. In addition, an *initial state distribution* is considered, with $p_0(s)$ denoting its probability (density) at $s$ for any $s \in \mathcal{S}$. A policy $\pi$ is the decision maker which observes a state $s$ and takes an action $a$ stochastically following a probability (density) of $\pi(a|s)$, at each time step. The interaction between $\pi$ and $\mathcal{M}$ induces a trajectory $\langle S_0, A_0, R_0 \rangle, \cdots, \langle S_t, A_t, R_t \rangle, \cdots$. Our work uses a decoder trained via generative adversarial networks (Goodfellow et al., 2014; Volz et al., 2018) to map a low-dimensional continuous latent vector to a game level segment. The action is a latent vector. The state is represented by a fixed number of latent vectors of recently generated segments. These latent vectors are concatenated into a single vector. Conventionally, the initial state is a randomly sampled latent vector. If there are not enough segments generated to construct a complete state, zeros will be padded into the vacant entries. The reward function is defined to evaluate the newly generated segment.

### 3.1 MULTIMODAL ENSEMBLE POLICY

Similar to (Ren et al., 2021), we use a multimodal ensemble policy that makes decisions following Gaussian mixture models. The ensemble policy $\pi$ consists of $m$ *sub-policies*, namely $\pi_1, \cdots, \pi_m$, and a *weight function* $\boldsymbol{\beta}(\cdot) : \mathcal{S} \mapsto \mathbb{R}^m$. The decision distribution determined by $\pi$ at state $s$, namely $\pi(\cdot|s)$, can be viewed as an $m$-component Gaussian mixture model. Each component $\pi_i(\cdot|s)$ is the decision distribution determined by $\pi_i$ at $s$, while its mixture weight $\beta_i(s)$ is given by the weight function with $\sum_{i=1}^{m} \beta_i(s) = 1$ holds. Specifically in this work, the components are i.i.d. spherical Gaussian distributions, and their means and standard deviations are denoted by $\boldsymbol{\mu}_i(s)$ and $\boldsymbol{\sigma}_i(s)$, i.e., $\pi_i(\cdot|s) = \mathcal{N}(\boldsymbol{\mu}_i(s), \boldsymbol{\sigma}_i(s)I)$. The ensemble policy samples a sub-policy $\pi_i$ based on the weight vector $\boldsymbol{\beta}(s)$ first, and then samples an action with the sub-policy $\pi_i$, i.e., the final output $a \sim \pi_i(\cdot|s)$.

We use $m + 1$ independent muti-layer perceptrons (MLPs) to model the ensemble policy. Each of the $m$ sub-policies is modelled by an *individual actor* using an MLP, while the weight function is modelled by a *selector* using an additional MLP. We regard the $m + 1$ MLPs as a union model with multiple output heads, though they do not share any common parameters. The union of their parameters is denoted by $\theta$. The $i^{\text{th}}$ individual actor outputs two vectors $\boldsymbol{\mu}_i^{\theta}(s)$ and $\boldsymbol{\sigma}_i^{\theta}(s)$, and the selector outputs an $m$-dimensional vector $\boldsymbol{\beta}^{\theta}(s)$ that represents the mixture weights.

### 3.2 NEGATIVE CORRELATION REGULARISATION FOR DIVERSITY

Inspired by (Liu & Yao, 1999), we propose a *negative correlation regularisation* to diversify the behaviours of sub-policies. The regularisation calculates the 2-Wasserstein distances (Olkin & Pukelsheim, 1982) $\omega_{i,j}(s)$ between each pair of Gaussian decision distributions $\pi_i(\cdot|s)$ and $\pi_j(\cdot|s)$. 2-Wasserstein distance is chosen because it is widely used and differentiable when both distributions are Gaussian. The formula of the distance measure is detailed in Section C.2. Let $\lceil \cdot \rceil^c$ denote a *down-clip function* bounding its argument under a constant upper bound $c$, the formulation of the negative correlation regularisation $\varrho(\cdot)$ is

$$\varrho(\pi(\cdot|s)) = \sum_{i=1}^{m} \sum_{j=1}^{m} \beta_i(s)\beta_j(s)\lceil \omega_{i,j}(s) \rceil^{\bar{\omega}}, \tag{1}$$

where $\bar{\omega}$ is a *clip size* parameter. This work arbitrarily sets $\bar{\omega} = 0.6\sqrt{d}$ where $d$ is the dimensionality of the action space. We abbreviate $\varrho(\pi(\cdot|s))$ as $\varrho^{\pi}(s)$ in the rest of this paper. Omitting the clipping function, $\varrho^{\pi}(s)$ is the expected Wasserstein distance of two sub-decision distributions stochastically sampled according to $\boldsymbol{\beta}(s)$. Two ideas motivate the use of the clipping function: (i) if the distance is already large, continuously maximising the distance does not benefit a lot for diversity of generated levels but harms the rewards; (ii) a few (even two) far-away clusters can have large expected distance but such pattern has limited diversity. Taking the down-clip helps with avoiding this case. $\varrho^{\pi}(s)$ is

maximised only if $\omega_{i,j}(s) \geq \bar{\omega}$ and $\beta_i(s) = \beta_j(s)$ for all pairs of $i$ and $j$. This regularisation term is integrated into the MDP, thus the objective to be maximised in NCERL is defined as

$$J^\pi = \mathbb{E}_{\mathcal{M},\pi} \left[ \sum\nolimits_{t=0}^{\infty} \gamma^t \big( r(S_t, A_t) + \lambda \varrho^\pi(S_t) \big) \right], \tag{2}$$

where $\lambda$ is a *regularisation coefficient* and $\gamma$ is the *discount rate*. The regularisation follows a different form compared to the reward $r(S_t, A_t)$, which is based on the decision distribution rather than a single action and is independent of the actual action taken by the policy. This raises the question of *how to optimise the regularised objective $J^\pi$?* We adapt the traditional RL theorems, *policy iteration* (Sutton & Barto, 2018) and *policy gradient* (Sutton et al., 1999) to answer it.

## 4 POLICY REGULARISATION THEOREMS

To answer the question above, this section derives *regularised policy iteration* and *policy-regularisation gradient*. All lemmas and theorems are proved in Appendix A.

**Value Functions** Similar to the standard one (Sutton & Barto, 2018), the state value for policy regularisation is defined as $V_\varrho^\pi(s) \doteq \mathbb{E}_{\mathcal{M},\pi} \left[ \sum_{k=0}^{\infty} \gamma^k \varrho^\pi(S_{t+k}) \mid S_t = s \right]$. The state-action value, however, is varied from the standard $Q$-value, defined as

$$Q_\varrho^\pi(s, a) \doteq \mathbb{E}_{\mathcal{M},\pi} \left[ \sum\nolimits_{k=1}^{\infty} \gamma^k \varrho^\pi(S_{t+k}) \,\Big|\, S_t = s, A_t = a \right]. \tag{3}$$

The counter $k$ starts from 1 rather than 0, because $\varrho^\pi$ is independent on the actual action. We further define a regularised state value function $\mathcal{V}_\varrho^\pi(s) = V^\pi(s) + \lambda V_\varrho^\pi(s)$ and a regularised state-action value function $\mathcal{Q}_\varrho^\pi(s, a) = Q^\pi(s, a) + \lambda Q_\varrho^\pi(s, a)$. An *optimal* policy $\pi^*$ is defined as $\forall s \in \mathcal{S}, \forall \pi \in \Pi, \mathcal{V}_\varrho^{\pi^*}(s) \geq \mathcal{V}_\varrho^\pi(s)$, where $\Pi$ is the hypothesis policy space. A $\pi^*$ maximises $J$ over $\Pi$.

### 4.1 REGULARISED POLICY ITERATION

We now describe the regularised policy iteration for an arbitrary policy regularisation $\varrho^\pi$. Off-policy regularised deep RL algorithms can be established by approximating this theoretical algorithm. Considering an RL algorithm in a tabular setting, we define a Bellman operator $\mathcal{T}_\varrho^\pi$ of $Q_\varrho$-function for all paired $s, a \in \mathcal{S} \times \mathcal{A}$ to derive the $Q_\rho$ of a policy as

$$\mathcal{T}_\varrho^\pi Q_\varrho(s, a) \leftarrow \mathbb{E}_{s' \sim p(\cdot|s,a), a' \sim \pi(\cdot|s')} \left[ \gamma \varrho^\pi(s') + \gamma Q_\varrho(s', a') \right]. \tag{4}$$

Having this definition, it is guaranteed that applying the operators over $\mathcal{S} \times \mathcal{A}$ repeatedly will converge to the true $Q_\varrho$-function of any policy $\pi$, as formalised below.

**Lemma 1** ($Q_\varrho$-Function Evaluation). *By repeatedly applying $Q_\varrho^{k+1} = \mathcal{T}_\varrho^\pi Q_\varrho^k$ from an arbitrary $Q_\varrho$-function $Q_\varrho^0$, the sequence $Q_\varrho^0, \cdots, Q_\varrho^k, \cdots$ converges to $Q_\varrho^\pi$ as $k \to \infty$.*

The complete regularised policy evaluation applies standard policy evaluation and $Q_\varrho$-function evaluation either jointly (by summing $Q^\pi$ with $\lambda Q_\varrho^\pi$ directly) or separately.

To derive an improved policy $\pi_{\text{new}}$ from an arbitrary policy $\pi_{\text{old}}$ assuming $\mathcal{Q}_\varrho^{\pi_{\text{old}}}(s, a)$ is known for any $s, a \in \mathcal{S} \times \mathcal{A}$, we define a greedy regularised policy improvement operator for all $s \in \mathcal{S}$ as

$$\pi_{\text{new}}(\cdot|s) \leftarrow \arg\max\nolimits_{\pi(\cdot|s) \in \Pi(\cdot|s)} \left[ \lambda \varrho^\pi(s) + \mathbb{E}_{a \sim \pi(\cdot|s)} \left[ \mathcal{Q}_\varrho^{\pi_{\text{old}}}(s, a) \right] \right]. \tag{5}$$

We say a policy $\pi'$ is *better* than another $\pi$, denoted as $\pi' \succ \pi$, if $\forall s \in \mathcal{S}, \mathcal{V}_\varrho^{\pi'}(s) \geq \mathcal{V}_\varrho^\pi(s)$ and $\exists s \in \mathcal{S}, \mathcal{V}_\varrho^{\pi'}(s) > \mathcal{V}_\varrho^\pi(s)$. Then a lemma of *regularised policy improvement* is formalised as follows.

**Lemma 2** (Regularised Policy Improvement). *For any $\pi_{\text{old}} \in \Pi$ and its $\pi_{\text{new}}$ derived via equation 5, it is guaranteed that $\pi_{\text{new}} \succ \pi_{\text{old}}$ if $\pi_{\text{old}}$ is not optimal.*

The regularised policy iteration algorithm alters between the regularised policy evaluation and the regularised policy improvement repeatedly. It is proved that given a finite $\Pi$, such an algorithm converges to an optimal policy over $\Pi$. This convergence guarantee is formulated into Theorem 1.

**Theorem 1** (Regularised Policy Iteration). *Given a finite hypothesis policy space $\Pi$, regularised policy iteration converges to an optimal policy over $\Pi$ from any $\pi_0 \in \Pi$.*

## 4.2 Policy-Regularisation Gradient

We derive the gradient for policy regularisation, namely the *policy-regularisation gradient* (PRG) to provide a theoretical foundation of regularised RL for on-policy algorithms.

An improper discounted state distribution $d^\pi$ is defined as $d^\pi(s) \doteq \sum_{t=0}^{\infty} \gamma^t \int_{\mathcal{S}} p_0(u) \mathbb{P}[u \xrightarrow{t} s, \pi] \, \mathrm{d}u$ like in standard policy gradient, where $\mathbb{P}[s \xrightarrow{k} s', \pi]$ denotes the probability density of transiting to $s'$ after $k$ steps from $s$, by applying $\pi$. Consider a policy represented by a parametric model $\pi^\theta$ where $\theta$ is its parameters. Using $\varrho^\theta(s)$ as the abbreviation of $\varrho(\pi^\theta(\cdot|s))$, PRG is formalised as follows.

**Theorem 2** (Policy-Regularisation Gradient, PRG). *The gradient of a policy regularisation objective* $J_\varrho^\theta = \mathbb{E}_{\mathcal{M}, \pi^\theta}[\sum_{t=0}^{\infty} \gamma^t \varrho^\theta(S_t)]$ *w.r.t. $\theta$ follows*

$$\frac{\partial J_\varrho^\theta}{\partial \theta} = \int_{\mathcal{S}} d^\pi(s) \left( \frac{\partial \varrho^\theta(s)}{\partial \theta} + \int_{\mathcal{A}} Q_\varrho^\pi(s, a) \frac{\partial \pi^\theta(a|s)}{\partial \theta} \, \mathrm{d}a \right) \mathrm{d}s = \mathbb{E}_{\substack{s \sim d^\pi, \\ a \sim \pi(\cdot|s)}} \left[ \frac{\partial \varrho^\theta(s)}{\partial \theta} + Q_\varrho^\pi(s, a) \frac{\partial \ln \pi^\theta(a|s)}{\partial \theta} \right]. \quad (6)$$

## 5 Implementing NCERL with Asynchronous Evaluation

### 5.1 Implementing NCERL Agent

We implement NCERL agents based on the soft-actor critic (SAC) (Haarnoja et al., 2018b). An NCERL agent carries two critics for soft $Q$-function, two critics for $Q_\varrho$-function and the ensemble policy. All critics use MLPs. With $Q_\varrho(s, a; \phi_1)$ and $Q_\varrho(s, a; \phi_2)$ denoting $Q_\varrho$-value predictions at state-action pair $s$, $a$ of two $Q_\varrho$-critics, where $\phi_1$ and $\phi_2$ denote their parameters, each critic $j \in \{1, 2\}$ is trained to minimise the Bellman residual of the negative correlation regularisation

$$\mathcal{L}_\phi = \mathbb{E}_{s,a,s' \sim \mathcal{D}} \left[ \left( Q_\varrho(s, a; \phi_j) - \gamma V_\varrho(s'; \bar{\phi}) \right)^2 \right], \quad (7)$$

where $\mathcal{D}$ is the replay memory. $V_\varrho(s'; \bar{\phi})$ is the prediction of $V_\varrho$-value with target parameters $\bar{\phi}$:

$$V_\varrho(s'; \bar{\phi}) = \varrho^\theta(s') + \sum_{i=1}^{m} \beta_i^\theta(s) \min_{j \in \{1,2\}} Q_\varrho(s', \tilde{a}_i'; \bar{\phi}_j), \quad (8)$$

where $\varrho^\theta(s')$ is the negative correlation regularisation (cf. equation 1) of the ensemble policy model $\pi^\theta$ at $s'$, $\tilde{a}_i'$ is an action sampled from $\pi_i^\theta(\cdot|s')$. The target parameters are updated with the same smoothing technique as in SAC. The $Q$-critics are trained with SAC, but when computing the soft $Q$-value target, we take expectation over all individual actors as in equation 8.

The ensemble policy model is updated by approximating the regularised policy improvement operator defined in equation 5 with critics and gradient ascent. The loss function is

$$\mathcal{L}_\theta = -\mathbb{E}_{s \sim \mathcal{D}} \left[ \lambda \varrho^\theta(s) + \sum_{i=1}^{m} \beta_i^\theta(s) \mathbb{E}_{\boldsymbol{\epsilon}_i \sim \mathcal{N}} \left[ \mathcal{Q}_\varrho(s, f^\theta(s, \boldsymbol{\epsilon}_i)) \right] \right], \quad (9)$$

where the reparametrisation trick is used with $f^\theta(s, \boldsymbol{\epsilon}_i) = \boldsymbol{\mu}_i^\theta(s) + \boldsymbol{\sigma}_i^\theta(s) \odot \boldsymbol{\epsilon}_i$ where $\boldsymbol{\epsilon}_i \sim \mathcal{N}(0, I)$, and the regularised state-action value is predicted by $\mathcal{Q}_\varrho(s, a) \leftarrow \min_{k \in \{1,2\}} Q(s, a; \psi_k) + \lambda \min_{j \in \{1,2\}} Q_\varrho(s, a; \phi_j)$. The gradient of $-\mathcal{L}_\theta$ is also an approximated estimation of PRG (cf. equation 6 and Appendix B). The agent learns through performing gradient descent using Adam optimiser (Kingma & Ba, 2015) periodically to $\mathcal{L}_\phi$ and $\mathcal{L}_\theta$, respectively. For exploration, the agent directly samples an action from the ensemble policy model.

### 5.2 Asynchronous Off-Policy Training Framework

Reward evaluation in OLG typically requires simulations of gameplay on complete levels to test their playability or simulate player behaviours (Wang et al., 2022). Such a simulation is usually realised by running a game-playing agent to play the level, which is time-consuming. The parallel off-policy training framework devised for OLG in previous work is synchronous (Wang et al., 2022), which causes unnecessary hang-on of CPU processes and GPU. On the other hand, existing asynchronous RL frameworks focus on on-policy algorithms (Mnih et al., 2016) or distributed learners (Gu et al., 2017), which do not apply to this work. Therefore, we design an asynchronous training framework. It separates model training and reward evaluation into two parallel processes, while the reward evaluation uses an asynchronous process pool to compute rewards of complete levels through multi-processing. This framework is also potentially beneficial to other tasks such as neural combinatorial optimisation where the reward evaluation operates on complete trajectories (Liu et al., 2023). Appendix C.1 provides the pseudo-code.

## 6    EXPERIMENTAL STUDIES

NCERL is evaluated on the well-known Mario level generation benchmark (Karakovskiy & To-gelius, 2012), with two different tasks. This section introduces the experiment settings and discusses experimental results. Appendix D.2 and E.1 provide more details and analysis.

### 6.1    EXPERIMENT SETTING

NCERL instances[1] with different numbers of individual actors and regularisation coefficient values are compared to several state-of-the-art policy ensemble algorithms and classic algorithms.

**Online Level Generation Tasks**    Algorithm instances are tested on two OLG tasks of a well-known benchmark (Hu et al., 2023) from recent literature, namely *MarioPuzzle* (Shu et al., 2021) and *MultiFacet* (Wang et al., 2022). They are mainly different in terms of the reward functions. *MarioPuzzle* defines two reward terms, *fun* and *historical deviation*. *Fun* restricts the divergence between new segments and old ones while *historical deviation* encourages the RL policy to generate novel segments in relation to previously generated ones. *MultiFacet* defines two reward terms to guide the RL policy to generate segments that introduce new tile patterns and play traces. Both of them use a *playability* reward, penalising the RL policy for generating unpassable segments. The full reward function in each task is a weighted sum of these reward terms. The two tasks are selected as they are the state-of-the-art for online Mario level generation and their source codes are publicly available online. Formulations and more details of the two tasks are presented in Appendix D.1.

**Compared Algorithms**    In total, 30 algorithm instances are used to train 5 independent generators each (thus 150 generators in total), which can be categorised into three groups. (i) 24 NCERL instances are trained with all the combinations of the ensemble size $m \in \{2, 3, 4, 5\}$ and regularisation coefficient $\lambda \in \{0.0, 0.1, 0.2, 0.3, 0.4, 0.5\}$. (ii) Three state-of-the-art ensemble RL algorithms published within the past three years, including PMOE (Ren et al., 2021), DvD (Parker-Holder et al., 2020) and SUNRISE (Lee et al., 2021) reviewed in Section 2. The algorithm proposed in (Yang et al., 2022) is excluded as it is designed for discrete action space. All of them are trained in our proposed asynchronous framework, using five individual actors following (Parker-Holder et al., 2020; Lee et al., 2021). During the test, one of the sub-policies is randomly selected to make the decision at each step for better diversity. (iii) Three SACs are trained in the standard single-process setting, in the synchronous multi-process framework in (Wang et al., 2022), and in our proposed asynchronous framework, referred to as SAC, EGSAC and ASAC, respectively.

All algorithm instances are trained for one million steps. Common hyperparameters shared by all instances are set to the same values. Hyperparameters uniquely belonging to an algorithm are set as reported in its original paper. Appendix D.2 reports all the hyperparameters and their values.

### 6.2    RESULTS AND DISCUSSION

**Performance Criteria**    All of the trained generators are tested by generating 500 levels of 25 segments each. Cumulative reward and diversity evaluated on the 500 levels generated by the generators are compared. The diversity is measured by the expectation of distance between pairs of generated levels, which is extensively used in PCG (Nam et al., 2021; Earle et al., 2021; Beukman et al., 2022). Additionally, geometric mean (G-mean) (Derringer, 1994) and average ranking (Avg-rank) (Zhang et al., 2022) are used to enable unified comparisons. The average ranking criterion estimates the average of reward rank and diversity rank of an algorithm instance out of all compared ones. Both unified criteria are suitable for combining multiple metrics in different scales. Appendix D.4 formulates the criteria and discusses more details.

**Effectiveness of NCERL**    Table 1 reports the performances of the tested algorithm instances. On both tasks, NCERL achieves the highest diversity. In terms of the G-mean criterion which unifies reward and diversity, NCERL almost outperforms all the other algorithms with any $\lambda$ except for PMOE. With $\lambda = 0.3$, $0.4$ and $0.5$, the G-mean score of NCERL is higher than PMOE on both tasks. The superior G-mean scores indicate that NCERL balances reward and diversity better and

---

[1]We refer to an algorithm with specific hyperparameters as an algorithm instance.

outperforms other compared algorithms in a unified sense. In terms of the other unified criterion, the average ranking, NCERL surpasses all other algorithms except for SUNRISE on the *MarioPuzzle* task. Compared to SUNRISE, NCERL allows one to make trade-offs between reward and diversity by specifying the regularisation coefficient. With $\lambda = 0.5$, NCERL achieves the best diversity score on both tasks. PMOE shows competitive performance in terms of diversity, which is only lower than the NCERL instance s of $\lambda = 0.2$ and $\lambda = 0.5$ on *MarioPuzzle* and only lower than the NCERL of $\lambda = 0.5$ on *MultiFacet*, but the reward gained by PMOE is worse than most NCERL instances. The reward gained by NCERL is not superior among all compared ones, but the enhancement to diversity is more significant than the sacrifice of reward, according to the results of the G-mean.

Table 1: Average and $\pm$standard deviation of each algorithm's performance over 5 independent trials. All ensemble algorithms reported in this table are trained with 5 individual actors.

| Task | Criterion | Non-ensemble | | | Ensemble ($m = 5$) | | | NCERL ($m = 5$) | | | | | |
|---|---|---|---|---|---|---|---|---|---|---|---|---|---|
| | | SAC | EGSAC | **ASAC** | PMOE | DvD | SUNRISE | $\lambda$=0.0 | $\lambda$=0.1 | $\lambda$=0.2 | $\lambda$=0.3 | $\lambda$=0.4 | $\lambda$=0.5 |
| Mario Puzzle | Reward | 56.21 ±.896 | 54.86 ±.966 | 57.22 ±1.00 | *46.39* ±*11.1* | 58.78 ±1.41 | **60.43** ±**1.08** | 55.24 ±2.20 | 51.42 ±10.9 | 53.78 ±3.72 | 53.22 ±7.15 | 54.59 ±2.68 | 53.26 ±3.23 |
| | Diversity | 628.5 ±173 | 809.9 ±77.1 | 760.8 ±123 | 1714 ±161 | 760.1 ±128 | 984.8 ±58.4 | 1342 ±326 | 1570 ±327 | 1940 ±101 | 1688 ±318 | 1698 ±262 | **1967** ±**278** |
| | G-mean | *186.1* ±*26.9* | 210.5 ±9.89 | 207.9 ±14.9 | 279.4 ±41.0 | 210.6 ±16.9 | 243.8 ±6.89 | 269.7 ±27.7 | 279.1 ±35.6 | **322.7** ±**14.5** | 296.1 ±19.4 | 302.8 ±19.3 | 322.1 ±19.3 |
| | Avg-rank | 8.32 ±.808 | *8.50* ±*.544* | 6.96 ±.450 | 7.14 ±.723 | 6.10 ±.829 | **4.28** ±**.319** | 6.16 ±.564 | 5.30 ±1.48 | 4.88 ±1.37 | 4.98 ±.778 | 5.54 ±1.03 | 5.04 ±.599 |
| Multi Facet | Reward | **46.93** ±**.270** | 44.35 ±.640 | 46.83 ±.456 | 37.40 ±8.59 | 45.68 ±2.29 | 46.53 ±.246 | 46.39 ±.376 | 46.16 ±.356 | 45.35 ±1.12 | 37.87 ±3.55 | 40.86 ±4.21 | *35.77* ±*5.52* |
| | Diversity | 249.6 ±22.7 | 451.6 ±62.7 | *213.0* ±*31.9* | 973.9 ±380 | 394.0 ±201 | 388.6 ±53.9 | 401.6 ±107 | 492.3 ±69.6 | 620.2 ±92.5 | 1024 ±151 | 889.6 ±234 | **1142** ±**224** |
| | G-mean | 108.1 ±4.94 | 141.2 ±10.3 | *99.56* ±*7.41* | 181.7 ±12.6 | 129.9 ±26.1 | 134.1 ±9.18 | 135.1 ±18.2 | 150.3 ±10.4 | 167.1 ±11.2 | 195.5 ±5.17 | 187.3 ±19.7 | **199.0** ±**6.94** |
| | Avg-rank | 6.14 ±.422 | *7.26* ±*.656* | 6.88 ±.752 | 6.10 ±.155 | 5.82 ±.640 | 5.76 ±.516 | 5.94 ±.408 | 5.64 ±.398 | **5.60** ±**.410** | 6.10 ±.0632 | 5.90 ±.237 | 6.06 ±.102 |

Figure 2 further shows the learning curves of tested algorithms. On both tasks, the reward of all algorithms ascends over time steps. Meanwhile, except for NCERL and PMOE, the diversity scores of all other algorithms descend over time steps. On the *MarioPuzzle* task, the G-mean scores of NCERL and PMOE ascend, while on *MultiFacet*, their G-mean values remain high. The G-mean values of all other algorithms descend on both tasks. As both NCERL and PMOE use multimodal policy, this observation implies that the multimodal policy is key to balancing reward and diversity. Overall, NCERL better balances reward and diversity.

To analyse the performance of each independent trial, we illustrate the locations of all trained generators in the reward-diversity objective space via scatter plots

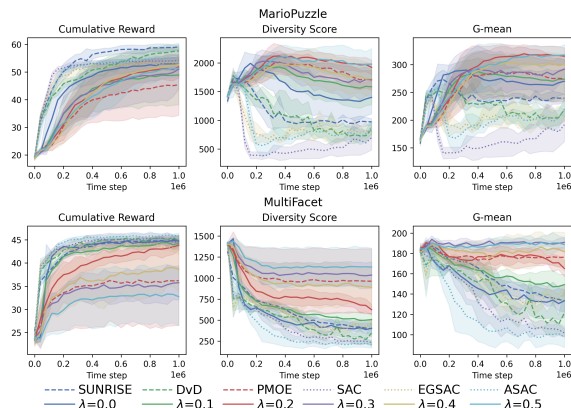

Figure 2: Solid curves correspond to the learning curves of NCERL instances with different values of $\lambda$. Dashed and dotted curves refer to the ones of compared algorithms. Shadow represents standard deviations over the five trials.

(Figure 3). According to Figure 3, the compared algorithms are generally located in regions of low diversity, while NCERLs spread widely and contribute a major portion of the Pareto front. Most of the generators trained by the compared algorithms are dominated by NCERL generators, while the non-dominated ones are generally biased towards the reward. The observations further indicate that NCERL is able to train generators with varied trade-offs between reward and diversity, making it possible to cater to varied preferences.

The locations of NCERL generators trained under the same hyperparameters sometimes vary a lot, revealing that NCERL can be instable over trials. Table 1 and Figure 2 also show the standard deviation of NCERL's performance is generally big. Meanwhile, PMOE also suffers from instability

across trials, especially on the *MultiFacet* task. As both PMOE and NCERL use multimodal policy, future work may investigate techniques to mitigate the instability of training multimodal policy. On the other hand, it is expected to integrate NCERL with multi-objective RL (Hayes et al., 2022), to train a set of non-dominated generators in which the instability may not be a disadvantage.

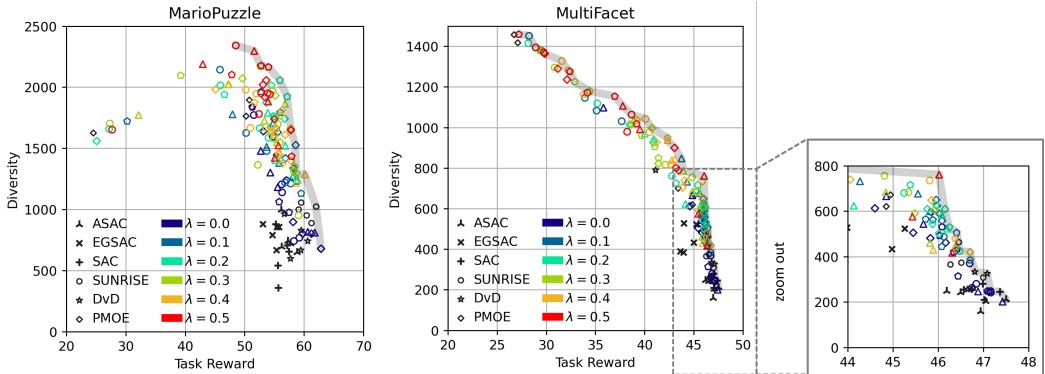

Figure 3: All trained generators' locations in the reward-diversity objective space are visualised in these scatter plots. Coloured markers correspond to generators trained with NCERL, where shapes represent the ensemble size $m$ (round: $m = 2$, triangle: $m = 3$, diamond: $m = 4$, pentagon: $m = 5$). Grey curves indicate the Pareto front across all trained generators.

**Verification of the Asynchronous Framework**     To verify our proposed asynchronous framework, SAC, EGSAC and ASAC are compared. According to Table 1, the EGSAC, which trains SAC in a synchronous framework (Wang et al., 2022), gets lower rewards and higher diversity scores. ASAC's performance is very similar to SAC, especially in terms of the cumulative reward. Therefore, it is more reliable to plug base algorithms into our proposed asynchronous framework. Meanwhile, our asynchronous framework is faster than the synchronous framework of EGSAC. We train generators with ASAC, EGSAC and standard SAC for one million time steps each on a computation platform with 64 CPU cores and GTX 2080 GPU to compare their time efficiency. Using 20 processes for evaluation, ASAC costs 4.93h while EGSAC costs 6.06h, i.e., ASAC is 22.9% faster than EGSAC. The standard single-process SAC costs 34.26h, i.e., ASAC speeds up SAC by 596%.

**Influence of Hypareparamters**     According to Table 1 and Figure 3, as $\lambda$ increases, the diversity score generally increases while the reward generally decreases. The influence of varying the ensemble size $m$ does not show clear regularities. We investigate and discuss the influence of ensemble size and regularisation coefficient more comprehensively in Appendix E.1.

## 7    CONCLUSION

In this paper, we propose a novel *negatively correlated ensemble RL* approach, to enable online diverse game level generation. The NCERL approach uses a Wasserstein distance-based regularisation to diversify the behaviour of a multimodal ensemble policy. Furthermore, an asynchronous off-policy training framework is designed to train online level generators faster. To show how the regularisation can be optimised in MDP, we derive the regularised RL theorems, which facilitate NCERL. NCERL is shown to be able to generate diverse game levels with competitive performance on the reward. It achieves superior G-mean scores, which indicates that NCERL better balances the reward and diversity. The proposed method and theorems make it possible to further develop multi-objective RL algorithms that consider the diversity of generated levels as an objective, which can train a set of non-dominated generators in one single trial to cater to varied preferences. Because the levels generated by NCERL are diverse, it is likely to enable fresh and interesting gameplay experiences even after numerous levels have been generated and played.

ACKNOWLEDGMENTS

This work was supported by the National Key R&D Program of China (Grant No. 2023YFE0106300), the National Natural Science Foundation of China (Grant No. 62250710682), the Shenzhen Science and Technology Program (Grant No. 20220815181327001), the Research Institute of Trustworthy Autonomous Systems, and the Guangdong Provincial Key Laboratory (Grant No. 2020B121201001).

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

## A   PROOFS

The proofs of Lemma 1, Lemma 2, Theorem 1 and Theorem 2 are provided in this section.

### A.1   BEHAVIOUR REGULARISED POLICY ITERATION

Our proof of regularised policy iteration follows the similar line of soft policy iteration (Haarnoja et al., 2018a) but considers a general decision distribution regularisation setting rather than the specific maximising entropy setting.

**Lemma 1** ($Q_\varrho$-Function Evaluation). *By repeatedly applying $Q_\varrho^{k+1} = \mathcal{T}_\varrho^\pi Q_\varrho^k$ from an arbitrary $Q_\varrho$-function $Q_\varrho^0$, the sequence $Q_\varrho^0, \cdots, Q_\varrho^k, \cdots$ converges to $Q_\varrho^\pi$ as $k \to \infty$.*

*Proof.* As the policy $\pi$ to be evaluated is fixed at this stage, we can treat $\gamma \mathbb{E}_{a \sim \pi(\cdot|s), s' \sim p(\cdot|s,a)}[\varrho^\pi(s')]$ as a reward function $\hat{r}(s, a)$, and then treat $Q_\varrho^\pi(s, a)$ as a $Q$-function since

$$Q_\varrho^\pi(s, a) \doteq \mathbb{E}_{\mathcal{M}, \pi} \left[ \sum\nolimits_{k=1}^\infty \gamma^k \varrho^\pi(S_{t+k}) \Big| S_t = s, A_t = a \right]$$

$$= \mathbb{E}_{\mathcal{M}, \pi} \left[ \sum\nolimits_{k=0}^\infty \gamma^k \left( \gamma \varrho^\pi(S_{t+k+1}) \right) \Big| S_t = s, A_t = a \right]$$

$$= \mathbb{E}_{\mathcal{M}, \pi} \left[ \sum\nolimits_{k=0}^\infty \gamma^k \hat{R}_{t+k} \Big| S_t = s, A_t = a \right],$$

where the last expression is the same as the definition of standard $Q$-function. Then we can simply borrow the theoretical results of standard policy evaluation (Sutton & Barto, 2018). □

**Lemma 2** (Regularised Policy Improvement). *For any $\pi_{\text{old}} \in \Pi$ and its $\pi_{\text{new}}$ derived via equation 5, it is guaranteed that $\pi_{\text{new}} \succ \pi_{\text{old}}$ if $\pi_{\text{old}}$ is not optimal.*

Recap of equation 5:

$$\forall s \in \mathcal{S}, \ \pi_{\text{new}}(\cdot|s) = \underset{\pi(\cdot|s) \in \Pi(\cdot|s)}{\arg \max} \left[ \lambda \varrho^\pi(s) + \mathbb{E}_{a \sim \pi(\cdot|s)} \left[ \mathcal{Q}_\varrho^{\pi_{\text{old}}}(s, a) \right] \right]. \tag{5}$$

*Proof.* By taking the operator described in equation 5, it is definite that $\forall s \in \mathcal{S}, \ \lambda \varrho^{\pi_{\text{new}}}(s) + \mathbb{E}_{a \sim \pi_{\text{new}}(\cdot|s)} \left[ \mathcal{Q}_\varrho^{\pi_{\text{old}}}(s, a) \right] \geq \lambda \varrho^{\pi_{\text{old}}}(s) + \mathbb{E}_{a \sim \pi_{\text{old}}(\cdot|s)} \left[ \mathcal{Q}_\varrho^{\pi_{\text{old}}}(s, a) \right] = \mathcal{V}_\varrho^{\pi_{\text{old}}}(s)$. With $\mathcal{Q}_\varrho^\pi(s, a) = r(s, a) + \mathbb{E}_{s' \sim p(\cdot|s,a)} \left[ \gamma \mathcal{V}^\pi(s') \right]$, we have:

$$\mathcal{V}^{\pi_{\text{old}}}(s) \leq \lambda \varrho^{\pi_{\text{new}}}(s) + \mathbb{E}_{a \sim \pi_{\text{new}}(\cdot|s)} \left[ \mathcal{Q}_\varrho^{\pi_{\text{old}}}(s, a) \right]$$

$$= \lambda \varrho^{\pi_{\text{new}}}(s) + \mathbb{E}_{a \sim \pi_{\text{new}}(\cdot|s)} \left[ r(s, a) + \mathbb{E}_{s' \sim p(\cdot|s,a)} \left[ \gamma \mathcal{V}_\varrho^{\pi_{\text{old}}}(s') \right] \right]$$

$$= \mathbb{E}_{a \sim \pi_{\text{new}}(\cdot|s)} [\lambda \varrho^{\pi_{\text{new}}}(s) + r(s, a)] + \gamma \mathbb{E}_{\mathcal{M}, \pi_{\text{new}}} \left[ \mathcal{V}_\varrho^{\pi_{\text{old}}}(S_{t+1}) \big| S_t = s \right]$$

$$\leq \mathbb{E}_{a \sim \pi_{\text{new}}(\cdot|s)} [\lambda \varrho^{\pi_{\text{new}}}(s) + r(s, a)] + \gamma \mathbb{E}_{\mathcal{M}, \pi_{\text{new}}} \left[ \lambda \varrho^{\pi_{\text{new}}}(S_{t+1}) + \mathbb{E}_{a' \sim \pi_{\text{new}}(\cdot|S_{t+1})} \left[ \mathcal{Q}_\varrho^{\pi_{\text{old}}}(S_{t+1}, a') \right] \Big| S_t = s \right]$$

$$= \sum\nolimits_{k=0}^1 \gamma^k \mathbb{E}_{\mathcal{M}, \pi_{\text{new}}} [\lambda \varrho^{\pi_{\text{new}}}(S_{t+k}) + R_{t+k}| S_t = s] + \gamma^2 \mathbb{E}_{\mathcal{M}, \pi_{\text{new}}} \left[ \mathcal{V}_\varrho^{\pi_{\text{old}}}(S_{t+2}) \big| S_t = s \right]$$

$$\vdots$$

$$\leq \mathbb{E}_{\mathcal{M}, \pi_{\text{new}}} \left[ \sum\nolimits_{k=0}^\infty \gamma^k (\lambda \varrho^{\pi_{\text{new}}}(S_{t+k}) + R_{t+k}) \right] = \mathcal{V}_\varrho^{\pi_{\text{new}}}(s).$$

Hence, $\forall s \in \mathcal{S}$, $\mathcal{V}_\varrho^{\pi_{\text{old}}}(s) = \mathcal{V}_\varrho^{\pi_{\text{new}}}(s)$ or $\pi_{\text{new}} \succ \pi_{\text{old}}$. If it is the former case, then $\forall s \in \mathcal{S}$ we have:

$$
\begin{aligned}
\mathcal{V}_\varrho^{\pi_{\text{new}}}(s) &= \max_{\pi \in \Pi} \left[ \lambda \varrho^\pi(s) + \mathbb{E}_{a \sim \pi(\cdot|s)}[\mathcal{Q}_\varrho^{\pi_{\text{new}}}(s,a)] \right] \\
&= \max_{\pi \in \Pi} \left[ \lambda \varrho^\pi(s) + \mathbb{E}_{a \sim \pi(\cdot|s)} \left[ r(s,a) + \gamma \mathbb{E}_{s' \sim p(\cdot|s,a)}[\mathcal{V}_\varrho^{\pi_{\text{new}}}(s')] \right] \right] \\
&= \max_{\pi \in \Pi} \mathbb{E}_{\mathcal{M},\pi} \left[ \lambda \varrho^\pi(S_t) + R_t + \gamma \mathcal{V}_\varrho^{\pi_{\text{new}}}(S_{t+1}) \big| S_t = s \right] \\
&= \max_{\pi \in \Pi} \mathbb{E}_{\mathcal{M},\pi} \left[ \sum_{k=0}^\infty \gamma^k \left( \lambda \varrho^\pi(S_{t+k}) + R_{t+k} \right) \big| S_t = s \right].
\end{aligned}
$$

It indicates that both $\pi_{\text{new}}$ and $\pi_{\text{old}}$ are optimal. Therefore, $\pi_{\text{new}} \succ \pi_{\text{old}}$ if $\pi_{\text{old}}$ is not optimal. □

**Theorem 1** (Regularised Policy Iteration). *Given a finite stochastic policy space $\Pi$, regularised policy iteration converges to an optimal policy over $\Pi$ from any $\pi_0 \in \Pi$.*

*Proof.* Collecting Lemma 1 and Lemma 2 with the theoretical result of standard $Q$-value evaluation (Sutton et al., 1999) and the condition that $\Pi$ is finite, evidently the policy converges to optimal. □

### A.2 STOCHASTIC POLICY GRADIENT FOR BEHAVIOUR REGULARISATION

Our proof for Theorem 2 draws lessons from the proof of stochastic policy gradient (Sutton et al., 1999) for basic ideas, and deterministic policy gradient (Silver et al., 2014) for dealing with continuous space. We alter the superscript between $\pi$ and $\theta$ and sometimes drop the $\theta$ term in the formulation of policy, such a denotation rule aims at emphasising whether the term requires gradient w.r.t. $\theta$ or not, but all the terms are induced by the parametric model $\pi^\theta(\cdot|\cdot)$.

We assume that the state space and action space are continuous, and the discounted regularisation objective is considered. Furthermore, we assume the involved functions are continuous over the space and have real number supremum whenever we exchange the orders of derivations and integrals.

**Theorem 2** (Policy-Regularisation Gradient, PRG). *The gradient of a policy regularisation objective $J_\varrho^\theta = \mathbb{E}_{\mathcal{M},\pi^\theta}[\sum_{t=0}^\infty \gamma^t \varrho^\theta(S_t)]$ w.r.t. $\theta$ follows*

$$
\begin{aligned}
\frac{\partial J_\varrho^\theta}{\partial \theta} &= \int_{\mathcal{S}} d^\pi(s) \left( \frac{\partial \varrho^\theta(s)}{\partial \theta} + \int_{\mathcal{A}} Q_\varrho^\pi(s,a) \frac{\partial \pi^\theta(a|s)}{\partial \theta} \, \mathrm{d}a \right) \mathrm{d}s \qquad (6) \\
&= \mathbb{E}_{\substack{s \sim d^\pi, \\ a \sim \pi(\cdot|s)}} \left[ \frac{\partial \varrho^\theta(s)}{\partial \theta} + Q_\varrho^\pi(s,a) \frac{\partial \ln \pi^\theta(a|s)}{\partial \theta} \right].
\end{aligned}
$$

*Proof.* According to the definition of the value functions of regularisation, we have

$$
V_\varrho^\pi(s) = \varrho^\pi(s) + \mathbb{E}_{a \sim \pi(\cdot|s)}[Q_\varrho^\pi(s,a)] = \varrho^\pi(s) + \int_{\mathcal{A}} \pi(a|s) Q_\varrho^\pi(s,a) \, \mathrm{d}a
$$

and

$$
Q_\varrho^\pi(s,a) = \mathbb{E}_{s' \sim p(\cdot|s,a)}[\gamma V_\varrho^\pi(s')] = \int_{\mathcal{S}} \gamma p(s'|s,a) V_\varrho^\pi(s') \, \mathrm{d}s'.
$$

Then we derive the bootstrap equation of $\dfrac{\partial V_\varrho^\theta(s)}{\partial \theta}$ as follows:

$$
\begin{aligned}
\frac{\partial V_\varrho^\theta(s)}{\partial \theta} &= \frac{\partial}{\partial \theta}\left[\varrho^\theta(s) + \int_{\mathcal{A}} \pi^\theta(a|s)Q_\varrho^\theta(s,a)\,\mathrm{d}a\right] \\
&= \frac{\partial \varrho^\theta(s)}{\partial \theta} + \int_{\mathcal{A}}\left(\frac{\partial \pi^\theta(a|s)}{\partial \theta}Q_\varrho^\pi(s,a) + \pi(a|s)\frac{\partial Q_\varrho^\theta(s,a)}{\partial \theta}\right)\,\mathrm{d}a \\
&= \frac{\partial \varrho^\theta(s)}{\partial \theta} + \int_{\mathcal{A}}\frac{\partial \pi^\theta(a|s)}{\partial \theta}Q_\varrho^\pi(s,a)\,\mathrm{d}a + \int_{\mathcal{A}}\pi(a|s)\frac{\partial}{\partial \theta}\int_{\mathcal{S}}\gamma p(s'|s,a)V_\varrho^\theta(s')\,\mathrm{d}s'\,\mathrm{d}a \\
&= \frac{\partial \varrho^\theta(s)}{\partial \theta} + \int_{\mathcal{A}}\frac{\partial \pi^\theta(a|s)}{\partial \theta}Q_\varrho^\pi(s,a)\,\mathrm{d}a + \int_{\mathcal{S}}\left(\gamma\int_{\mathcal{A}}\pi(a|s)p(s'|s,a)\,\mathrm{d}a\right)\frac{\partial V_\varrho^\theta(s')}{\partial \theta}\,\mathrm{d}s' \\
&= \frac{\partial \varrho^\theta(s)}{\partial \theta} + \int_{\mathcal{A}}\frac{\partial \pi^\theta(a|s)}{\partial \theta}Q_\varrho^\pi(s,a)\,\mathrm{d}a + \int_{\mathcal{S}}\gamma\mathbb{P}[s\xrightarrow{1}s',\pi]\frac{\partial V_\varrho^\theta(s')}{\partial \theta}\,\mathrm{d}s'.
\end{aligned}
\tag{10}
$$

Note $\int_{\mathcal{S}}\mathbb{P}[s\xrightarrow{t}u,\pi]\int_{\mathcal{S}}\mathbb{P}[u\xrightarrow{1}s',\pi]f(s')\,\mathrm{d}s'\,\mathrm{d}u = \int_{\mathcal{S}}\mathbb{P}[s\xrightarrow{t+1}s',\pi]f(s')\,\mathrm{d}s'$ is held for any $t>0$ as the Markov property is satisfied by any MDP. So we can unroll the result of equation 10 as follows:

$$
\frac{\partial V_\varrho^\theta(s)}{\partial \theta} = \sum_{t=0}^{\infty}\int_{\mathcal{S}}\gamma^t\mathbb{P}[s\xrightarrow{t}s',\pi]\left(\frac{\partial \varrho^\theta(s')}{\partial \theta} + \int_{\mathcal{A}}\frac{\partial \pi^\theta(a|s')}{\partial \theta}Q_\varrho^\pi(s',a)\,\mathrm{d}a\right)\,\mathrm{d}s',
\tag{11}
$$

where the item of $t=0$ is an improper integral that represents $\dfrac{\partial \varrho^\theta(s)}{\partial \theta} + \int_{\mathcal{A}}\dfrac{\partial \pi^\theta(a|s)}{\partial \theta}Q_\varrho^\pi(s,a)\,\mathrm{d}a$.

As $J_\varrho^\theta = \mathbb{E}_{\mathcal{M},\pi}\left[\sum_{t=0}^{\infty}\gamma^t\varrho^\theta(S_t)\right] = \int_{\mathcal{S}}p_0(s)V_\varrho^\theta(s)\,\mathrm{d}s$, together with equation 11 we have

$$
\begin{aligned}
\frac{\partial J_\varrho^\theta}{\partial \theta} &= \frac{\partial}{\partial \theta}\int_{\mathcal{S}}p_0(s)V_\varrho^\theta(s)\,\mathrm{d}s = \int_{\mathcal{S}}p_0(s)\frac{\partial V_\varrho^\theta(s)}{\partial \theta}\,\mathrm{d}s \\
&= \int_{\mathcal{S}}p_0(s)\sum_{t=0}^{\infty}\left(\int_{\mathcal{S}}\gamma^t\mathbb{P}[s\xrightarrow{t}s',\pi]\left(\frac{\partial \varrho^\theta(s')}{\partial \theta} + \int_{\mathcal{A}}\frac{\partial \pi^\theta(a|s')}{\partial \theta}Q_\varrho^\pi(s',a)\,\mathrm{d}a\right)\,\mathrm{d}s'\right)\,\mathrm{d}s \\
&= \sum_{t=0}^{\infty}\int_{\mathcal{S}}\int_{\mathcal{S}}p_0(s)\gamma^t\mathbb{P}[s\xrightarrow{t}s',\pi]\left(\frac{\partial \varrho^\theta(s')}{\partial \theta} + \int_{\mathcal{A}}\frac{\partial \pi^\theta(a|s')}{\partial \theta}Q_\varrho^\pi(s',a)\,\mathrm{d}a\right)\,\mathrm{d}s\,\mathrm{d}s' \\
&= \int_{\mathcal{S}}\left(\sum_{t=0}^{\infty}\int_{\mathcal{S}}\gamma^t p_0(s)\mathbb{P}[s\xrightarrow{t}s',\pi]\,\mathrm{d}s\right)\left(\frac{\partial \varrho^\theta(s')}{\partial \theta} + \int_{\mathcal{A}}\frac{\partial \pi^\theta(a|s')}{\partial \theta}Q_\varrho^\pi(s',a)\,\mathrm{d}a\right)\,\mathrm{d}s' \\
&= \int_{\mathcal{S}}d^\pi(s')\left(\frac{\partial \varrho^\theta(s')}{\partial \theta} + \int_{\mathcal{A}}Q_\varrho^\pi(s',a)\frac{\partial \pi^\theta(a|s')}{\partial \theta}\,\mathrm{d}a\right)\,\mathrm{d}s'.
\end{aligned}
\tag{12}
$$

For the $\int_{\mathcal{A}}Q_\varrho^\pi(s',a)\dfrac{\partial \pi^\theta(a|s')}{\partial \theta}\,\mathrm{d}a$ term, we can apply the log-derivative trick as follows:

$$
\begin{aligned}
\int_{\mathcal{A}}Q_\varrho^\pi(s',a)\frac{\partial \pi^\theta(a|s')}{\partial \theta}\,\mathrm{d}a &= \int_{\mathcal{A}}Q_\varrho^\pi(s',a)\pi(a|s')\frac{\partial \pi^\theta(a|s')}{\pi(a|s')\partial \theta}\,\mathrm{d}a \\
&= \int_{\mathcal{A}}\pi(a|s')Q_\varrho^\pi(s',a)\frac{\partial \ln \pi^\theta(a|s')}{\partial \theta}\,\mathrm{d}a \\
&= \mathbb{E}_{a\sim\pi(\cdot|s')}\left[Q_\varrho^\pi(s',a)\frac{\partial \ln \pi^\theta(a|s')}{\partial \theta}\right].
\end{aligned}
\tag{13}
$$

Concluding equation 12 and equation 13, we get equation 6. $\qquad\square$

# B    EXPLAINING ENTROPY REGULARISED RL WITH GENERAL POLICY REGULARISATION THEOREMS

The soft policy iteration (Haarnoja et al., 2018b) improves a policy $\pi_{\text{old}}$ by

$$\pi_{\text{new}}(\cdot|s) = \arg\min_{\pi \in \Pi} D_{\text{KL}}\left(\pi(\cdot|s) \left\| \frac{\exp(\mathcal{Q}_{\mathcal{H}}^{\pi_{\text{old}}}(s,\cdot)/\alpha)}{Z^{\pi_{\text{old}}}(s)}\right.\right).$$

This is a specific case of equation 5 with $\varrho^\pi(s) = \mathcal{H}(\pi(\cdot|s))$ and $\lambda = \alpha$, where $\mathcal{H}$ denotes the entropy, since

$$
\begin{aligned}
&\arg\min_{\pi \in \Pi} D_{\text{KL}}\left(\pi(\cdot|s) \left\| \frac{\exp(\mathcal{Q}_{\mathcal{H}}^{\pi_{\text{old}}}(s,\cdot)/\lambda)}{Z^{\pi_{\text{old}}}(s)}\right.\right) \\
&= \arg\min_{\pi \in \Pi}\left[\int_{\mathcal{A}} \pi(a|s) \log \frac{\pi(a|s) Z^{\pi_{\text{old}}}(s)}{\exp(\mathcal{Q}_{\mathcal{H}}^{\pi_{\text{old}}}(s,a)/\lambda)}\,\mathrm{d}a\right] \\
&= \arg\min_{\pi \in \Pi}\left[\int_{\mathcal{A}} \pi(a|s) \log \pi(a|s)\,\mathrm{d}a + \int_{\mathcal{A}} \pi(a|s) \log \frac{1}{\exp(\mathcal{Q}_{\mathcal{H}}^{\pi_{\text{old}}}(s,a)/\lambda)}\,\mathrm{d}a + \int_{\mathcal{A}} \pi(a|s) \log Z^{\pi_{\text{old}}}(s)\,\mathrm{d}a\right] \\
&= \arg\min_{\pi \in \Pi}\left[-\mathcal{H}(\pi(\cdot|s)) - \frac{1}{\lambda}\int_{\mathcal{A}} \pi(a|s)\mathcal{Q}_{\mathcal{H}}^{\pi_{\text{old}}}(s,a)\,\mathrm{d}a\right] \\
&= \arg\max_{\pi \in \Pi}\left[\lambda\varrho^\pi(s) + \mathbb{E}_{a\sim\pi(\cdot|s)}[\mathcal{Q}_{\mathcal{H}}^{\pi_{\text{old}}}(s,a)]\right].
\end{aligned}
$$

At the same time, the gradient of actor loss of SAC, can be viewed as a weighted summation of standard SPG and PRG in terms of entropy. The actor loss of SAC (Haarnoja et al., 2018b) is written as

$$J_\theta = \mathbb{E}_{s\sim\mathcal{D}, a\sim\pi^\theta(\cdot|s)}\left[\alpha \log(\pi^\theta(\cdot|s)) - \mathcal{Q}_{\mathcal{H}}(s,a)\right].$$

We borrow idea from (Degris et al., 2012), consider a behavioural policy $\varpi$ so that $s \sim d^\varpi \equiv s \sim \mathcal{D}$, then we have

$$
\begin{aligned}
J_\theta &= \mathbb{E}_{s\sim\mathcal{D}, a\sim\pi^\theta(\cdot|s)}\left[\alpha \log(\pi^\theta(\cdot|s)) - \mathcal{Q}_{\mathcal{H}}(s,a)\right] \\
&= \mathbb{E}_{s\sim d^\varpi}\left[-\mathcal{H}^\theta(s) - \int_{\mathcal{A}} \pi^\theta(a|s)(Q(s,a) + \lambda Q_{\mathcal{H}}(s,a))\,\mathrm{d}a\right] \\
-\frac{\partial J_\theta}{\partial\theta} &= \underbrace{\mathbb{E}_{s\sim d^\varpi}\left[\int_{\mathcal{A}} \frac{\partial\pi^\theta(a|s)}{\partial\theta}Q(s,a)\,\mathrm{d}a\right]}_{\text{Standard SPG}} + \underbrace{\lambda\mathbb{E}_{s\sim d^\varpi}\left[\frac{\partial\varrho^\theta(s)}{\partial\theta} + \int_{\mathcal{A}} \frac{\partial\pi^\theta(a|s)}{\partial\theta}Q_{\mathcal{H}}(s,a)\,\mathrm{d}a\right]}_{\text{PRG for entropy}}.
\end{aligned}
$$

Similarly, by $s \sim d^\varpi \equiv s \sim \mathcal{D}$, we can explain NCERL as optimising a weighted summation of behavioural SPG and behavioural PRG.

# C    ADDITIONAL DETAILS

## C.1    ASYNCHRONOUS OFF-POLICY TRAINING FRAMEWORK

We employ an evaluation pool to enable asynchronous evaluation. The pool contains a capacited queue and multiple processes. Once a task is submitted, the pool assigns the task to a free process if available, otherwise, the task will be stored in the queue temporarily. If the queue is full, the pool will be blocked until any process is finished. Algorithm 1 describes our asynchronous framework.

---

**Algorithm 1** Main Procedure of the Asynchronous Off-Policy Training Framework.

---

**Require:** Number of evaluation workers $w$, horizon of MDP $h$, update interval itv
 1: Create a multi-processing evaluation pool with $w$ workers
 2: credits $\leftarrow 0$
 3: **repeat**
 4:     Sample a no-reward trajectory $\vec{z}$ via the agent
 5:     Submit $\tau$ to the evaluation pool
 6:     $T \leftarrow$ collect evaluated trajectories from the pool
 7:     Update $\mathcal{D}$ with $T$
 8:     credits $\leftarrow$ credits $+ \frac{|T|}{\text{itv}}$
 9:     **for** $u : 1 \rightarrow \min\{\lfloor\text{credits}\rfloor, \lceil\frac{1.25wh}{\text{itv}}\rceil\}$ **do**
10:         $\mathcal{B} \leftarrow$ sample a batch from $\mathcal{D}$
11:         Update agent with $\mathcal{B}$
12:         credits $\leftarrow$ credits $- 1$
13:     **end for**
14: **until** run out of interaction budget
15: Waiting for all remaining tasks finished
16: $T \leftarrow$ collect all trajectories from the pool
17: Update $\mathcal{D}$ with $T$
18: **for** $u : 1 \rightarrow$ credits $+ \frac{|T|}{\text{itv}}$ **do**
19:     $\mathcal{B} \leftarrow$ sample a batch from $\mathcal{D}$
20:     Update agent with $\mathcal{B}$ via the base algorithm
21: **end for**

---

The coefficient 1.25 in line 9 of Algorithm 1 is arbitrarily set. We just arbitrarily assign a number slightly higher than 1, aiming to submit the simulation tasks uniformly, which reduces the suspends of simulation workers. The coefficient should be larger than 1 so that the credits can be used up.

### C.2  2-Wasserstein Distance

The 2-Wasserstein distance for a pair of probability distributions $X$ and $Y$ is defined as

$$\omega(X, Y) = \left( \inf_{Z \in \mathcal{Z}(X,Y)} \mathbb{E}_{(x,y) \sim Z}\big[\|x - y\|_2\big] \right)^{\frac{1}{2}},$$

where $Z$ is a joint distribution of $X$ and $Y$ and $\mathcal{Z}$ is the set of all joint distribution of $X$ and $Y$.

In case both $X$ and $Y$ are Gaussian, the 2-Wasserstein distance can be expressed as follows.

$$\omega^2(\mathcal{N}_1, \mathcal{N}_2) = \|\boldsymbol{\mu}_1 - \boldsymbol{\mu}_2\|_2^2 + \text{Trace}\left( \Sigma_1 + \Sigma_2 - 2\big(\Sigma_2^{\frac{1}{2}} \Sigma_1 \Sigma_2^{\frac{1}{2}}\big)^{\frac{1}{2}} \right),$$

where $\mathcal{N}_1, \mathcal{N}_2$ are the two Gaussian distribution to be compared, $\boldsymbol{\mu}_1$ and $\boldsymbol{\mu}_2$ are the means of $\mathcal{N}_1, \mathcal{N}_2$, and $\Sigma_1, \Sigma_2$ are the covariance matrices of $\mathcal{N}_1, \mathcal{N}_2$, respectively.

## D  Additional Experiment Details

### D.1  Online Level Generation Tasks

In our tasks, the state space is a $dn$-dimensional continuous vector space, where $d$ is the dimensionality of the latent vector of the action decoder and $n$ is the number of recently generated segments considered in the reward function. A state is a concatenated vector of a fixed number of latent vectors of recently generated segments. If there are not enough segments have been generated ($< n$) to construct a state, zeros will be padded in the vacant entries. The action space is a $d$-dimensional continuous vector space. An action is a latent vector which can be decoded into a level segment by the decoder. The decoder is a trained GAN in this work.

The reward functions of the two tasks considered in this paper consist of several reward terms. Those reward terms are described and formulated as follows.

**Playability**   The work of (Shu et al., 2021) and (Wang et al., 2022) use different formulation of playability. We use the one of (Wang et al., 2022) as it is the most recent one. Formally, the playability reward is

$$P(x_t) = \begin{cases} 0, & \text{if } x_{t-1} \oplus x_t \text{ is playable,} \\ -1, & \text{otherwise,} \end{cases}$$

where $\oplus$ represents appending a level segment with another. The playability is judged by the strongest game-playing agent in the Mario-AI-Framework benchmark (Karakovskiy & Togelius, 2012).

**Fun**   The fun reward (Shu et al., 2021) uses four configuration parameters lb, ub, $\delta$ and $n$. Furthermore, a metric $\text{TPKL}(\cdot, \cdot)$ is used to measure the dissimilarity of levels. Formally, the fun reward is

$$F(x_t) = \begin{cases} -(\bar{D}(x_t) - \text{lb})^2, & \text{if } \bar{D}(x_t) < \text{lb,} \\ -(\bar{D}(x_t) - \text{ub})^2, & \text{if } \bar{D}(x_t) > \text{ub,} \\ 0, & \text{otherwise,} \end{cases}$$

where

$$\bar{D}(x_t) = \frac{1}{n+1} \sum_{i=0}^{n} \text{TPKL}(x_t, \text{SW}(x_t, i\delta)),$$

where $\text{SW}(x_t, i\delta)$ represents the level segment extracted by sliding a window from $x_t$ backward with a stride of $i\delta$ tiles.

**Historical Deviation**   Historical deviation (Shu et al., 2021) uses two configuration parameters $m$ and $n$ with $m > n$. Formally, the historical deviation reward is

$$H(x_t) = \frac{1}{n} \min_X \sum_{x' \in X} \text{TPKL}(x_t, x'),$$
$$\text{s.t. } X \subset \{x_{t-m}, \cdots, x_{t-1}\} \wedge |X| = n.$$

**Level Novelty**   Level novelty (Wang et al., 2022) uses two configuration parameters $g$ and $n$. Furthermore, a metric $\text{TPJS}(\cdot, \cdot)$ is used to measure the dissimilarity of levels. Formally, the level novelty reward function is

$$L(x_t) = \frac{\sum_{i=1}^{n} \text{AC}\big(\text{TPJS}(x_t, x_{t-i}); g, r_i\big)}{\sum_{i=1}^{n} r_i},$$

in which $r_i = 1 - \frac{i}{n+1}$ and

$$\text{AC}(u; g, r) = \min\left\{r, 1 - \frac{|u - g|}{g}\right\}.$$

**Gameplay Novelty**   Gameplay novelty (Wang et al., 2022) shares the same form with the level novelty but replaces the TPJS metric with another metric $D_G(\cdot, \cdot)$ evaluating the distance between the simulated gameplay trace of two levels or level segments. Let $\text{gp}(x)$ be simulated gameplay of arbitrary level segment $x$, formulation of gameplay novelty is

$$G(x_t) = \frac{\sum_{i=1}^{n} \text{AC}\big(D_G(\text{gp}(x_t), \text{gp}(x_{t-i})); g, r_i\big)}{\sum_{i=1}^{n} r_i}.$$

All the configuration parameters are set as suggested in the corresponding papers and are summarised in Table 2.

The two tasks use weighted sums of their proposed reward terms as the final reward function. The observation space varies over different tasks since different reward functions depend on different numbers of latest level segments. Table 3 summarises the information of six tasks tested in this paper.

Table 2: Configuration parameters of the reward functions.

| Indicator | F | | | | H | | L | | G | |
|---|---|---|---|---|---|---|---|---|---|---|
| Parameter | lb | ub | $\delta$ | $n$ | $m$ | $n$ | $g$ | $n$ | $g$ | $n$ |
| Value | 0.26 | 0.94 | 8 | 21 | 10 | 5 | 0.3 | 5 | 0.14 | 5 |

Table 3: Summary of the OLG tasks.

| Task | Reward Function | Obsevation Space |
|---|---|---|
| MarioPuzzle | $R_t = 30\mathrm{F}(x_t) + 3\mathrm{H}(x_t) + 3\mathrm{P}(x_t)$ | $10d = 200$ |
| MultiFacet | $R_t = \mathrm{L}(x_t) + \mathrm{G}(x_t) + \mathrm{P}(x_t)$ | $5d = 100$ |

## D.2 HYPERPARAMETERS

The hyperparameters are listed in Table 4.

Using a discounted rate at $0.9$ which is not close to 1 is counter-intuitive as it can induce a large bias to the optimal policy in the average reward criterion being considered in OLG. However, we consider the OLG tasks satisfying a *perfectible* property. In this case, using any $\gamma > 0$ does not bias the optimal policy, so we can use a relatively small $\gamma$ which reduces the variance of gradient estimation. The next subsection details the assumed property.

## D.3 PERFECTIBLE PROPERTY AND SETTING OF DISCOUNT FACTOR

Two criteria are typically considered in RL, namely the *average reward criterion* $J_{\mathrm{A}}(\pi) = \lim_{h \to \infty} \mathbb{E}_{\mathcal{M},\pi} \left[ \frac{1}{h} \sum_{t=0}^{h-1} R_t \right]$ and the *discounted reward criterion* $J_D(\pi) = \mathbb{E}_{\mathcal{M},\pi} \left[ \sum_{t=0}^{\infty} \gamma^t R_t \right]$, where $\gamma \in [0,1]$ is the discount factor. In principle, the average reward criterion should be considered in OLG. However, we found that in some OLG tasks, relatively small $\gamma$ (e.g., 0.9) ensures a superior average reward despite that it may make $J_D$ badly approximate $J_A$.

To explain the aforementioned phenomenon, we assume that the MDP of some OLG tasks permits *perfect* policy after investigating the formulations of reward function in OLG (see Appendix D.1 for the reward functions). A perfect policy always gains the maximum reward. An MDP is said to be *perfectible* if it permits perfect policy. Let $\mathcal{S}^{\pi}$ be the set of all the states that possibly appear at any time step given a policy $\pi$, and $r^*$ be the maximum value of reward $r(s,a)$ over $\mathcal{S} \times \mathcal{A}$, the assumption is formalised as follows.

**Assumption 1** (Perfectible MDP). *For the MDP of OLG tasks considered in this work, there exists a perfect policy $\pi^{\circ}$ that satisfies $\forall s \in \mathcal{S}^{\pi^{\circ}}$, $\mathbb{P}[r(s,a) = r^* \mid a \sim \pi^{\circ}(\cdot|s)] = 1$.*

Under Assumption 1, we have the following proposition.

**Proposition 1.** *For a perfectible MDP, any optimal policy in terms of $J_{\mathrm{A}}$ and any optimal policy in terms of $J_{\mathrm{D}}$ (with $0 < \gamma \leq 1$) are all perfect policies.*

The proof of Proposition 1 is straightforward. If an optimal policy $\pi^*$ is not perfect, then there must be $\exists s \in \mathcal{S}^{\pi^{\circ}}, V^{\pi^{\circ}}(s) > V^{\pi^*}(s)$, in terms of both $J_A$ and $J_D$ criteria. This is in contrast to that $\pi^*$ is optimal. Therefore, we can use a $\gamma$ that is not close to 1 to optimise $J_A$ without bias of optimal policy in perfectible MDP. Intuitively, smaller $\gamma$ can reduce the variance, while larger $\gamma$ is likely to induce fewer local optima. One may need to set $\gamma$ carefully to enable superior average return in perfectible MDP.

## D.4 PERFORMANCE CRITERIA

Let $n = 500$ and $h = 25$ be the number of levels generated by each generator for the test and the number of segments in each level ($h = 25$ is used because this is slightly longer than the longest level in the training level set of the GAN.), the performance criteria are described below.

Table 4: Hyperparameter settings.

| Hyperparameter | Value |
|---|---|
| Optimiser (all networks and $\alpha$) | Adam Kingma & Ba (2015) |
| Learning rate (all networks and $\alpha$) | $3.0 \times 10^{-4}$ |
| Hidden layer activation (all networks) | ReLU |
| Number of hidden layers (all networks) | 2 |
| Size of hidden layer(all networks) | 256 |
| Batch size | 256 |
| Replay buffer size | $5 \times 10^5$ |
| Target smoothing coefficient | 0.02 |
| Target entropy | $-\dim(\mathcal{A}) = -20$ |
| Discount factor | 0.9 |
| Number of evaluation workers | 20 |
| Size of waiting queue | 25 |
| Update interval (see Algorithm 1) | 2 |

**Cumulative Reward**  Cumulative reward for a generator is calculated as $R = \sum_{t=1}^{h} R_t$ for each level, i.e., each MDP trajectory, then averaged over the 500 levels.

**Diversity Score**  Diversity score of a generator is calculated as

$$D = \frac{2}{n(n-1)} \sum_{i=1}^{n} \sum_{j \neq i} \Delta(x_i, x_j)$$

, where $\Delta(\cdot, \cdot)$ indicates the Hamming distance, i.e., how many different tiles are there between the two levels to be compared; and $x_i$, $x_j$ indicate the $i^{\text{th}}$ one and the $j^{\text{th}}$ one in the $n$ levels.

**Geometric Mean**  G-Mean for a generator is calculated as $G = \sqrt{RD}$. It is suitable to combine $R$ and $D$ even though they are in different scales. Because given any scaling coefficients $s_R > 0$ and $s_D > 0$ to rescale the cumulative reward and diversity score, the ratio between any two generators' G-mean values is constant since

$$\frac{G_1'}{G_2'} = \frac{\sqrt{s_R R_1 s_D D_1}}{\sqrt{s_R R_1 s_D D_1}} = \frac{\sqrt{R_1 D_1}}{\sqrt{R_1 D_1}} = \frac{G_1}{G_2},$$

where the subscripts 1 and 2 indicate the two generators being compared in terms of G-mean.

**Average Ranking**  For the average ranking, we first rank the 60 generators ($K = 12$ algorithm instances being compared $\times$ $T = 5$ independent trials) in terms of reward and diversity, respectively, from the highest to the lowest. With $k_R$ and $k_D$ denoting the ranks of a generator in terms of reward and diversity out of the $KT$ generators, respectively, the average ranking of this generator is calculated as $A = \frac{1}{2}(k_R + k_D)/T$.

## E  ADDITIONAL EXPERIMENT RESULTS

### E.1  INFLUENCE OF HYPAREPARAMTERS

We add tables 5 and 6 to show the performance of each independent NCERL generator to better analyse the training results, especially the effect of varying hyperparameters.

Table 5: Reward and diversity of all NCERL generators trained on MarioPuzzle. Trials 1–5 indicate the five independent training trials. In each row, the five trials are sorted according to the reward of trained generators.

| $m$ | $\lambda$ | Trial 1 | | Trial 2 | | Trial 3 | | Trial 4 | | Trial 5 | |
|---|---|---|---|---|---|---|---|---|---|---|---|
| | | Reward | Diversity | Reward | Diversity | Reward | Diversity | Reward | Diversity | Reward | Diversity |
| 2 | 0.0 | 59.21 | 769.8 | 58.28 | 743.1 | 56.26 | 1141 | 56.24 | 1206 | 51.54 | 1775 |
| | 0.1 | 57.71 | 1213 | 56.79 | 1221 | 53.68 | 1481 | 50.24 | 1626 | 45.81 | 2144 |
| | 0.2 | 55.11 | 1762 | 53.99 | 1588 | 52.53 | 1664 | 45.92 | 2017 | 27.21 | 1660 |
| | 0.3 | 59.07 | 950.8 | 54.85 | 1580 | 52.62 | 2171 | 52.21 | 1366 | 27.31 | 1706 |
| | 0.4 | 55.64 | 1450 | 54.96 | 1614 | 53.53 | 1676 | 50.94 | 1668 | 50.16 | 1980 |
| | 0.5 | 54.25 | 1943 | 52.75 | 1952 | 52.39 | 1784 | 48.47 | 2344 | 27.68 | 1654 |
| 3 | 0.0 | 61.76 | 811.5 | 60.62 | 818.7 | 55.52 | 1184 | 54.31 | 1304 | 52.72 | 1479 |
| | 0.1 | 58.15 | 1272 | 55.76 | 1380 | 54.70 | 1703 | 53.65 | 1513 | 47.91 | 1779 |
| | 0.2 | 58.96 | 1242 | 56.82 | 1522 | 56.06 | 1412 | 55.19 | 1406 | 53.88 | 1792 |
| | 0.3 | 56.79 | 1867 | 55.43 | 1548 | 53.31 | 1729 | 47.19 | 2026 | 32.10 | 1772 |
| | 0.4 | 60.13 | 1288 | 55.85 | 1471 | 55.22 | 1681 | 54.37 | 1644 | 47.33 | 2025 |
| | 0.5 | 55.63 | 1523 | 55.02 | 1424 | 53.90 | 1884 | 51.56 | 2298 | 42.95 | 2189 |
| 4 | 0.0 | 62.85 | 680.0 | 61.23 | 802.3 | 59.71 | 803.7 | 58.11 | 898.9 | 53.11 | 1642 |
| | 0.1 | 58.53 | 1528 | 58.21 | 1340 | 56.83 | 1419 | 55.53 | 1631 | 55.22 | 1665 |
| | 0.2 | 58.35 | 1270 | 56.73 | 1743 | 54.35 | 1675 | 53.61 | 1876 | 25.11 | 1562 |
| | 0.3 | 58.67 | 1295 | 56.02 | 1769 | 55.99 | 1551 | 55.16 | 1931 | 49.61 | 2074 |
| | 0.4 | 57.79 | 1400 | 57.75 | 1664 | 56.77 | 1612 | 56.36 | 1377 | 45.09 | 1984 |
| | 0.5 | 57.76 | 1650 | 55.01 | 1739 | 53.62 | 2058 | 53.36 | 1920 | 53.05 | 2022 |
| 5 | 0.0 | 57.46 | 976.5 | 57.00 | 1240 | 55.80 | 1063 | 54.66 | 1593 | 51.29 | 1839 |
| | 0.1 | 59.49 | 1133 | 58.69 | 1232 | 55.97 | 1794 | 52.74 | 1966 | 30.23 | 1723 |
| | 0.2 | 57.15 | 1925 | 55.89 | 2057 | 54.80 | 1764 | 54.47 | 2015 | 46.59 | 1940 |
| | 0.3 | 58.41 | 1351 | 58.20 | 1313 | 55.62 | 1702 | 54.66 | 1974 | 39.23 | 2098 |
| | 0.4 | 58.73 | 1212 | 56.20 | 1808 | 54.50 | 1648 | 51.88 | 1946 | 51.63 | 1874 |
| | 0.5 | 57.89 | 1434 | 53.99 | 2167 | 53.86 | 1954 | 52.76 | 2178 | 47.82 | 2102 |

According to the table, the generators with larger $m$ seem to perform more stable since there are more bad generators found in the group of $m = 2$. By increasing $\lambda$, the diversity is generally improved while the reward is generally decreased. However, the change in performance is not monotonic. The reason may be summed up as that the regularisation is not identical to the diversity score and it also promotes the exploration, making the effect of $\lambda$ not fully predictable. Reward and diversity are generally conflicted, but there are examples that a generator performs better than another in terms of both reward and diversity, i.e., dominate another. For example, Trial 2 of $m = 2, \lambda = 0.5$ dominates Trial 3 of $m = 2, \lambda = 0.5$. Some bad generators fail to gain good rewards while their diversity scores are not superior either (e.g., Trial 5 of $m = 2, \lambda = 0.2$, Trial 5 of $m = 2, \lambda = 0.3$, Trial 5 of $m = 3, \lambda = 0.3$). That means NCERL is not totally stable. Probably the regularisation objective sometimes leads to some local optima during training.

Table 6: Reward and diversity of all NCERL generators trained on MultiFacet. Trials 1–5 indicate the five independent training trials. In each row, the five trials are sorted according to the reward of trained generators.

| $m$ | $\lambda$ | Trial 1 | | Trial 2 | | Trial 3 | | Trial 4 | | Trial 5 | |
| --- | --- | --- | --- | --- | --- | --- | --- | --- | --- | --- | --- |
| | | Reward | Diversity | Reward | Diversity | Reward | Diversity | Reward | Diversity | Reward | Diversity |
| 2 | 0.0 | 47.16 | 242.7 | 47.15 | 248.4 | 46.84 | 280.2 | 46.74 | 263.7 | 46.69 | 370.6 |
| | 0.1 | 46.12 | 487.5 | 37.65 | 1031 | 35.11 | 1084 | 33.91 | 1147 | 28.23 | 1452 |
| | 0.2 | 46.46 | 424.5 | 45.24 | 681.1 | 42.77 | 761.4 | 35.16 | 1120 | 28.16 | 1417 |
| | 0.3 | 45.34 | 684.4 | 44.81 | 755.3 | 42.10 | 818.4 | 30.82 | 1296 | 29.48 | 1382 |
| | 0.4 | 45.80 | 735.1 | 40.48 | 978.3 | 40.07 | 1043 | 32.23 | 1282 | 31.56 | 1329 |
| | 0.5 | 38.65 | 1063 | 38.21 | 978.7 | 34.16 | 1172 | 29.73 | 1371 | 28.90 | 1396 |
| 3 | 0.0 | 47.42 | 201.7 | 46.88 | 246.8 | 45.67 | 543.9 | 44.85 | 666.7 | 35.79 | 1099 |
| | 0.1 | 46.38 | 444.2 | 45.81 | 600.9 | 45.55 | 677.9 | 44.27 | 732.3 | 43.73 | 848.1 |
| | 0.2 | 46.19 | 523.9 | 46.07 | 583.5 | 44.13 | 623.7 | 40.91 | 945.7 | 39.97 | 971.0 |
| | 0.3 | 45.81 | 461.3 | 44.84 | 684.1 | 43.93 | 787.5 | 38.88 | 1035 | 29.41 | 1382 |
| | 0.4 | 46.37 | 445.5 | 45.88 | 430.1 | 43.07 | 844.6 | 42.74 | 839.7 | 42.32 | 937.3 |
| | 0.5 | 46.31 | 417.3 | 46.01 | 761.3 | 45.43 | 575.6 | 39.49 | 992.5 | 37.77 | 1108 |
| 4 | 0.0 | 47.11 | 245.2 | 46.16 | 461.7 | 45.82 | 481.6 | 45.44 | 505.2 | 44.60 | 613.4 |
| | 0.1 | 46.08 | 558.2 | 46.02 | 589.8 | 45.97 | 624.8 | 45.95 | 649.3 | 45.85 | 595.4 |
| | 0.2 | 46.14 | 609.7 | 45.87 | 543.8 | 45.76 | 610.0 | 40.87 | 926.5 | 38.46 | 1016 |
| | 0.3 | 45.92 | 606.7 | 45.48 | 591.1 | 41.40 | 822.7 | 40.73 | 998.0 | 38.67 | 1016 |
| | 0.4 | 46.70 | 420.5 | 45.84 | 648.3 | 44.05 | 739.1 | 43.85 | 704.7 | 41.18 | 927.7 |
| | 0.5 | 43.42 | 789.2 | 43.03 | 900.4 | 32.09 | 1237 | 31.15 | 1290 | 29.84 | 1364 |
| 5 | 0.0 | 47.08 | 248.3 | 46.43 | 314.3 | 46.33 | 420.9 | 46.08 | 531.3 | 46.02 | 493.4 |
| | 0.1 | 46.70 | 384.7 | 46.40 | 506.1 | 46.07 | 444.1 | 45.88 | 561.8 | 45.72 | 565.0 |
| | 0.2 | 46.24 | 529.9 | 46.01 | 501.4 | 45.95 | 627.5 | 45.38 | 716.5 | 43.18 | 725.8 |
| | 0.3 | 41.38 | 849.7 | 41.04 | 905.6 | 39.68 | 960.1 | 34.38 | 1180 | 32.87 | 1226 |
| | 0.4 | 46.37 | 480.5 | 42.66 | 817.9 | 42.31 | 948.8 | 39.20 | 1037 | 33.77 | 1163 |
| | 0.5 | 43.18 | 802.0 | 39.15 | 1018 | 36.94 | 1154 | 32.39 | 1276 | 27.21 | 1460 |

The observation of this table is similar to the Table 5, the generators with larger $m$ seem to perform more stable while $\lambda$ is positively correlated to diversity but negatively correlated to reward. The diversity of those generators is generally smaller than the ones trained on MarioPuzzle. That means the reward function of MultiFacet may not allow super highly-diverse generators.

### E.2 INDIVIDUAL ACTOR SELECTION PROBABILITY

To see whether each of the sub-policy is used, we report the selection probability of each sub-policy for two NCERL generators trained with $\lambda = 0.5$ and $\lambda = 0.1$, in Tables 7 and 8.

Table 7: Selection probability at each step in two stochastic generation trials with the same initial state. An NCERL generator trained with $\lambda = 0.5$ and $m = 5$ on MarioPuzzle is picked to showcase. Levels generated in these two trials are presented in Figure 4. Bold text indicates the sub-policy of the corresponding row is selected at the corresponding time step. The probabilities smaller than 0.001 are notated as $\approx 0$.

| $\lambda = 0.5$ | | $t=1$ | $t=2$ | $t=3$ | $t=4$ | $t=5$ | $t=6$ | $t=7$ | $t=8$ | $t=9$ | $t=10$ | $t=11$ | $t=12$ | $t=13$ | $t=14$ | $t=15$ |
|---|---|---|---|---|---|---|---|---|---|---|---|---|---|---|---|---|
| | $\beta_1$ | $\approx 0$ | .062 | .084 | .138 | .094 | .147 | .147 | .144 | .171 | .137 | .179 | .184 | **.208** | .206 | .209 |
| | $\beta_2$ | .357 | **.260** | .242 | .218 | **.238** | .218 | .222 | .225 | .205 | .241 | .209 | **.209** | .188 | .189 | .190 |
| Run 1 | $\beta_3$ | **.239** | .216 | .225 | .233 | .247 | .222 | .233 | **.236** | .211 | **.191** | .200 | .208 | .195 | .211 | .195 |
| | $\beta_4$ | .404 | .259 | **.254** | .196 | .237 | .219 | **.205** | .203 | **.201** | .259 | .211 | .188 | .196 | **.178** | **.197** |
| | $\beta_5$ | $\approx 0$ | .202 | .195 | **.216** | .184 | **.193** | .194 | .192 | .212 | .172 | **.201** | .211 | .212 | .216 | .208 |
| | $\beta_1$ | $\approx 0$ | .035 | .074 | .159 | .132 | .149 | .149 | .140 | .164 | **.169** | .178 | **.198** | **.155** | .120 | .203 |
| | $\beta_2$ | **.357** | **.272** | .264 | .212 | .233 | .219 | .218 | .226 | .214 | .203 | .201 | .192 | .213 | .259 | **.191** |
| Run 2 | $\beta_3$ | .239 | .192 | .214 | .218 | .241 | .228 | .222 | .231 | .195 | .212 | .218 | .200 | .218 | .218 | .198 |
| | $\beta_4$ | .404 | .285 | .276 | **.210** | .202 | .209 | .208 | **.204** | **.227** | .209 | .191 | .199 | .216 | **.278** | .198 |
| | $\beta_5$ | $\approx 0$ | .216 | **.173** | .202 | **.193** | **.196** | **.203** | .199 | .199 | .207 | **.213** | .211 | .198 | .125 | .210 |

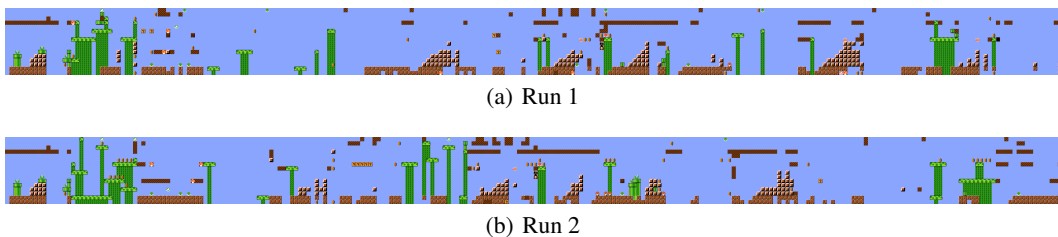

(a) Run 1

(b) Run 2

Figure 4: Generated levels of the two generation trials illustrated in Table 7.

Table 7 shows that the selection probability of the generator trained with $\lambda = 0.5$ is adjusted adaptively during the generation process, and all the sub-policies are used within the two generation trials.

Table 8: Selection probability at each step in two stochastic generation trials with the same initial state. An NCERL generator trained with $\lambda = 0.1$ and $m = 5$ on MarioPuzzle is picked to showcase. Levels generated in these two trials are presented in Figure 5. Bold text indicates the sub-policy of the corresponding row is selected at the corresponding time step. The probabilities smaller than 0.001 are notated as $\approx 0$.

| $\lambda = 0.1$ | | $t=1$ | $t=2$ | $t=3$ | $t=4$ | $t=5$ | $t=6$ | $t=7$ | $t=8$ | $t=9$ | $t=10$ | $t=11$ | $t=12$ | $t=13$ | $t=14$ | $t=15$ |
|---|---|---|---|---|---|---|---|---|---|---|---|---|---|---|---|---|
| | $\beta_1$ | **1.00** | .382 | **.531** | .501 | **.609** | .522 | .483 | **.539** | .560 | **.513** | **.610** | .711 | **.661** | **.687** | .522 |
| | $\beta_2$ | $\approx 0$ | $\approx 0$ | $\approx 0$ | $\approx 0$ | $\approx 0$ | $\approx 0$ | $\approx 0$ | $\approx 0$ | $\approx 0$ | $\approx 0$ | $\approx 0$ | $\approx 0$ | $\approx 0$ | $\approx 0$ | .009 |
| Run 1 | $\beta_3$ | $\approx 0$ | $\approx 0$ | $\approx 0$ | $\approx 0$ | $\approx 0$ | $\approx 0$ | $\approx 0$ | $\approx 0$ | .003 | $\approx 0$ | .016 | .005 | .003 | $\approx 0$ | .035 |
| | $\beta_4$ | $\approx 0$ | **.618** | .469 | **.499** | .391 | **.478** | **.517** | .461 | **.437** | .487 | .370 | **.284** | .336 | .313 | .420 |
| | $\beta_5$ | $\approx 0$ | $\approx 0$ | $\approx 0$ | $\approx 0$ | $\approx 0$ | $\approx 0$ | $\approx 0$ | $\approx 0$ | $\approx 0$ | $\approx 0$ | .003 | $\approx 0$ | $\approx 0$ | $\approx 0$ | .015 |
| | $\beta_1$ | **1.00** | .435 | .529 | .502 | **.502** | **.450** | .430 | **.534** | **.502** | **.470** | .453 | .350 | **.506** | **.474** | .400 |
| | $\beta_2$ | $\approx 0$ | $\approx 0$ | $\approx 0$ | $\approx 0$ | $\approx 0$ | $\approx 0$ | $\approx 0$ | $\approx 0$ | $\approx 0$ | $\approx 0$ | .012 | $\approx 0$ | $\approx 0$ | $\approx 0$ | .012 |
| Run 2 | $\beta_3$ | $\approx 0$ | $\approx 0$ | $\approx 0$ | $\approx 0$ | $\approx 0$ | $\approx 0$ | $\approx 0$ | $\approx 0$ | $\approx 0$ | $\approx 0$ | .046 | $\approx 0$ | .013 | $\approx 0$ | .026 |
| | $\beta_4$ | $\approx 0$ | **.565** | **.471** | **.498** | .498 | .550 | **.570** | .466 | .498 | .530 | **.468** | **.650** | .479 | .526 | .546 |
| | $\beta_5$ | $\approx 0$ | $\approx 0$ | $\approx 0$ | $\approx 0$ | $\approx 0$ | $\approx 0$ | $\approx 0$ | $\approx 0$ | $\approx 0$ | $\approx 0$ | .021 | $\approx 0$ | .002 | $\approx 0$ | **.016** |

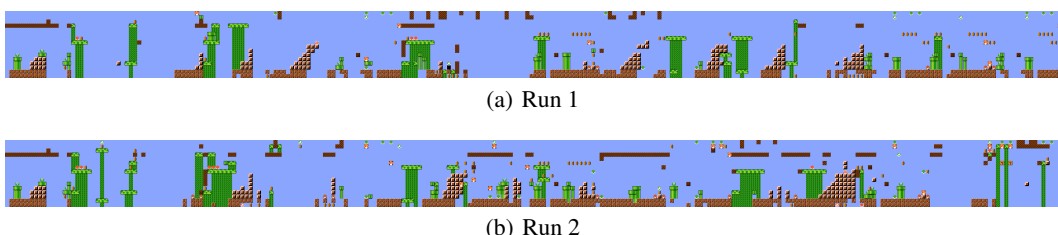

(a) Run 1

(b) Run 2

Figure 5: Generated levels of the two generation trials illustrated in Table 8.

Table 8 shows that some of the selection probability is near zero. Sub-policies 2 and 3 are never used within the two trials. This is because $\lambda$ is small.

### E.3 GENERATED SAMPLES

The following pages show partial examples generated by several trained NCERL generators, with a comparison to the examples generated by a standard SAC. Performance in terms of reward and diversity is reported. Our anonymous code repository[2] includes generated examples of all the generators we trained in this work.

---

[2] https://anonymous.4open.science/r/NCERL-Diverse-PCG-4F25/

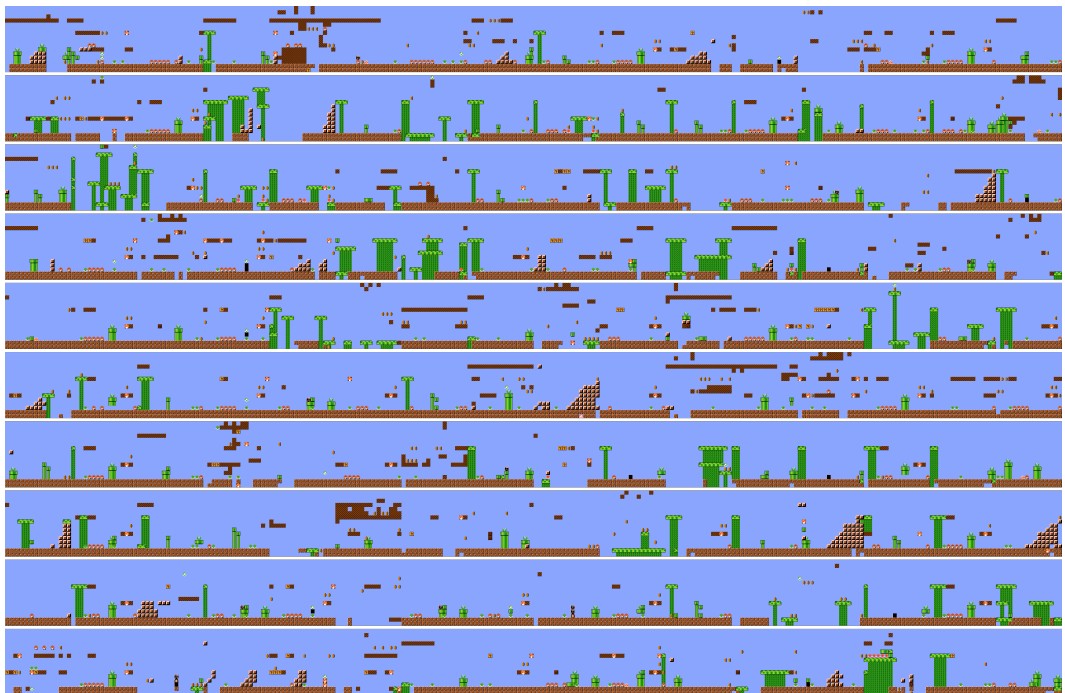

(a) Example levels generated by an NCERL generator trained with $\lambda = 0.5, m = 5$ on the MultiFacet task. The reward and diversity scores of this generator are $36.9$ and $1154$, respectively.

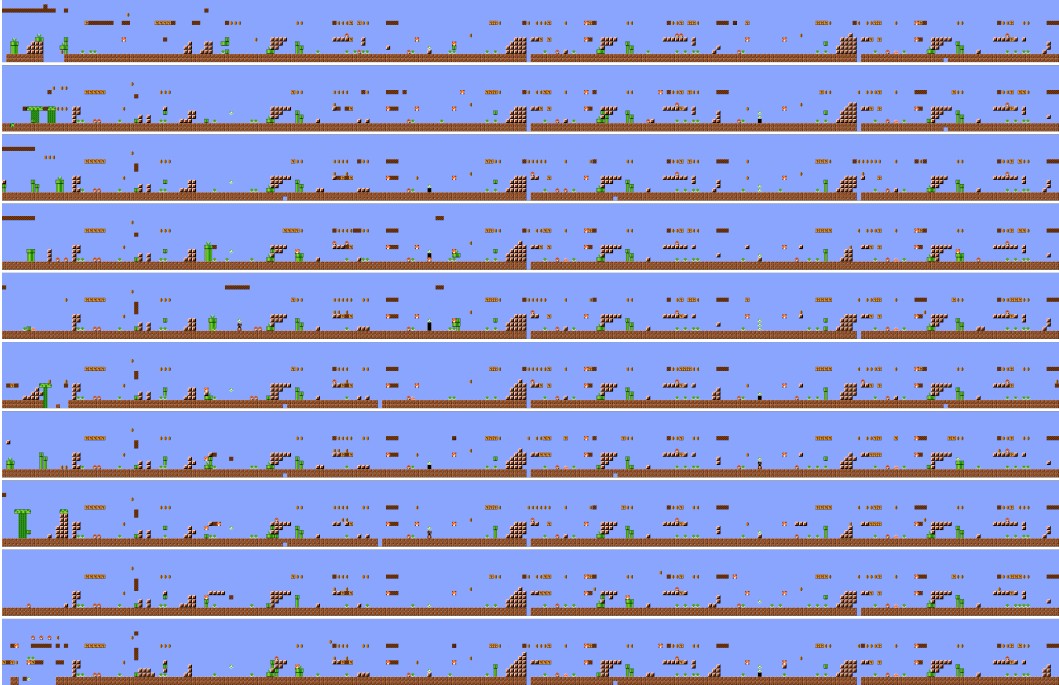

(b) Example levels generated by a SAC generator trained on the MultiFacet task. The reward and diversity scores of this generator are $46.7$ and $256.1$, respectively.

Figure 6: Example levels generated by an NCERL generator trained on MultiFacet with $\lambda = 0.5, m = 5$ and a SAC generator. SAC generated similar levels while NCERL generated diverse levels.

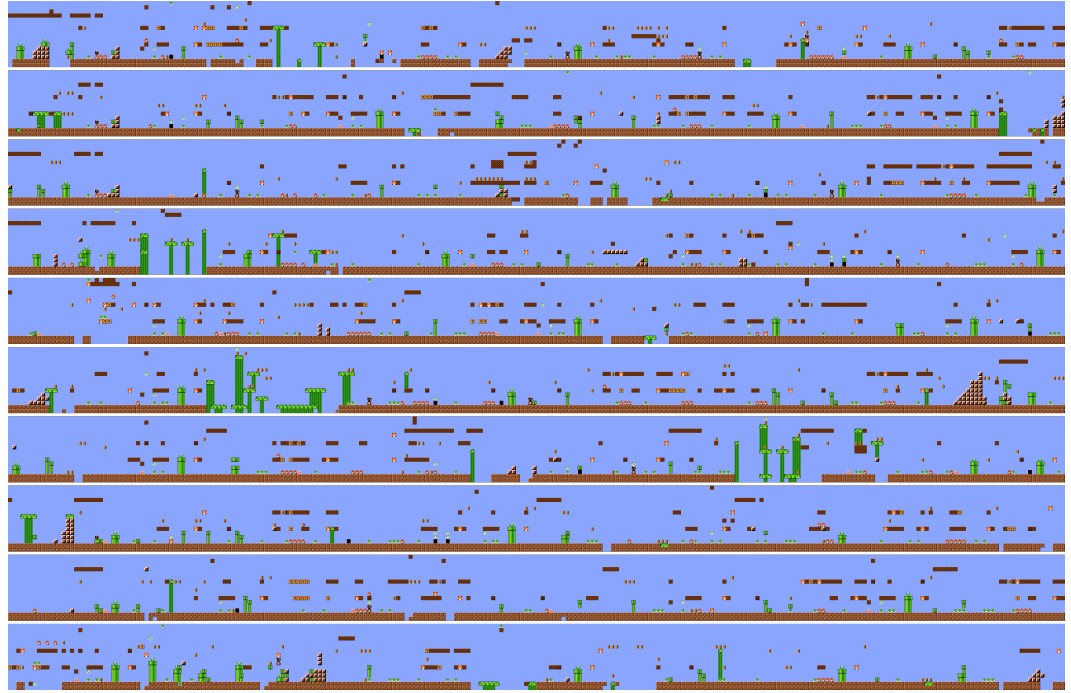

(a) Example levels generated by an NCERL generator trained with $\lambda = 0.3, m = 5$ on the MultiFacet task. The reward and diversity scores of this generator are $39.7$ and $960.1$, respectively.

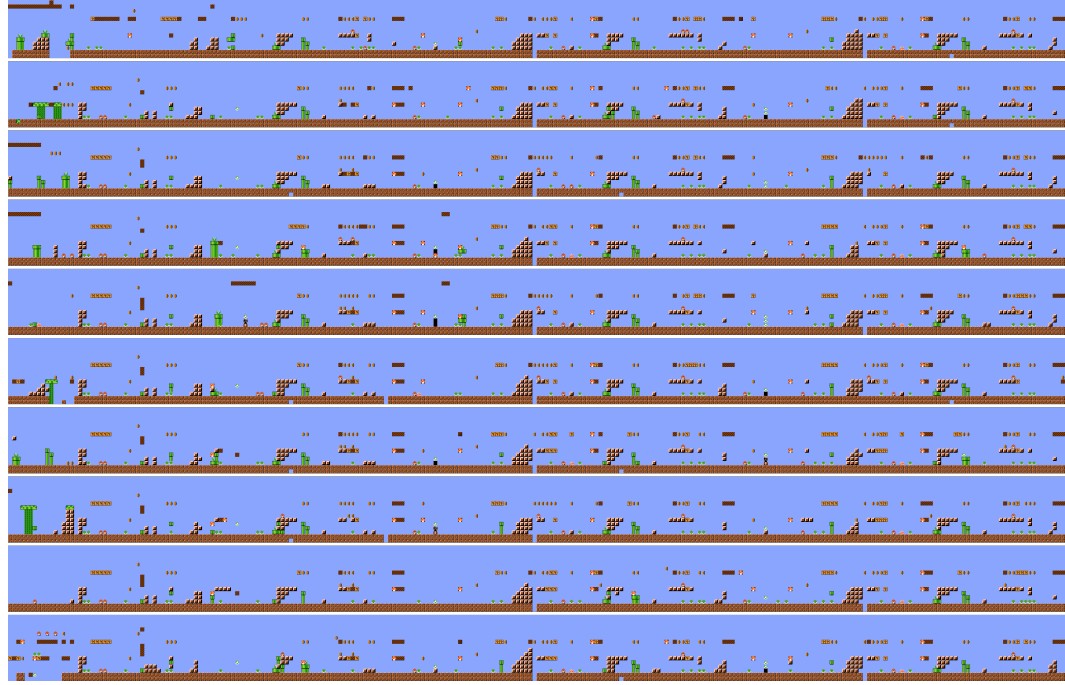

(b) Example levels generated by a SAC generator trained on the MultiFacet task. The reward and diversity scores of this generator are $46.7$ and $256.1$, respectively.

Figure 7: Example levels generated by an NCERL generator trained on MultiFacet with $\lambda = 0.3, m = 5$ and a SAC generator.

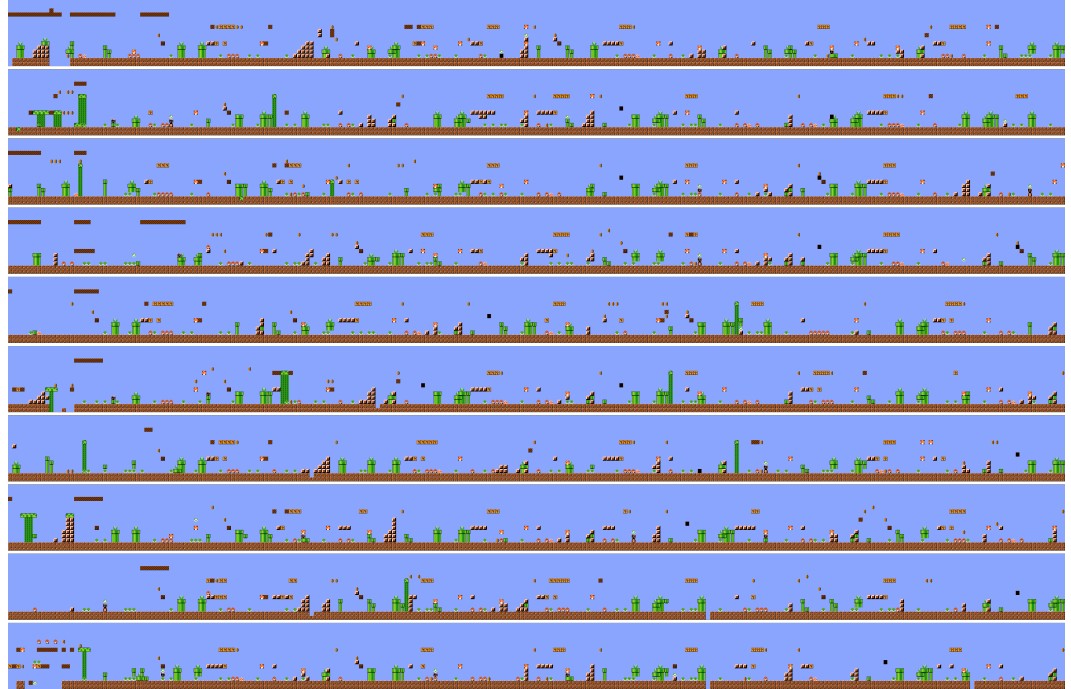

(a) Example levels generated by an NCERL generator trained with $\lambda = 0.1, m = 5$ on the MultiFacet task. The reward and diversity scores of this generator are $45.9$ and $561.8$, respectively.

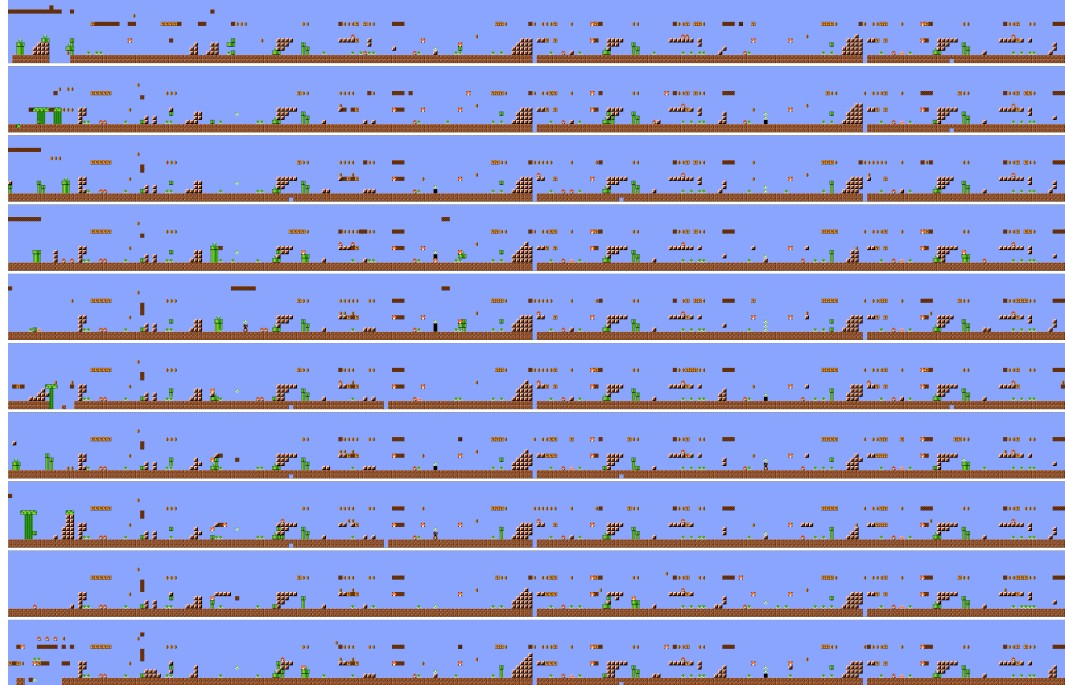

(b) Example levels generated by a SAC generator trained on the MultiFacet task. The reward and diversity scores of this generator are $46.7$ and $256.1$, respectively.

Figure 8: Example levels generated by an NCERL generator trained on MultiFacet with $\lambda = 0.1, m = 5$ and a SAC generator.

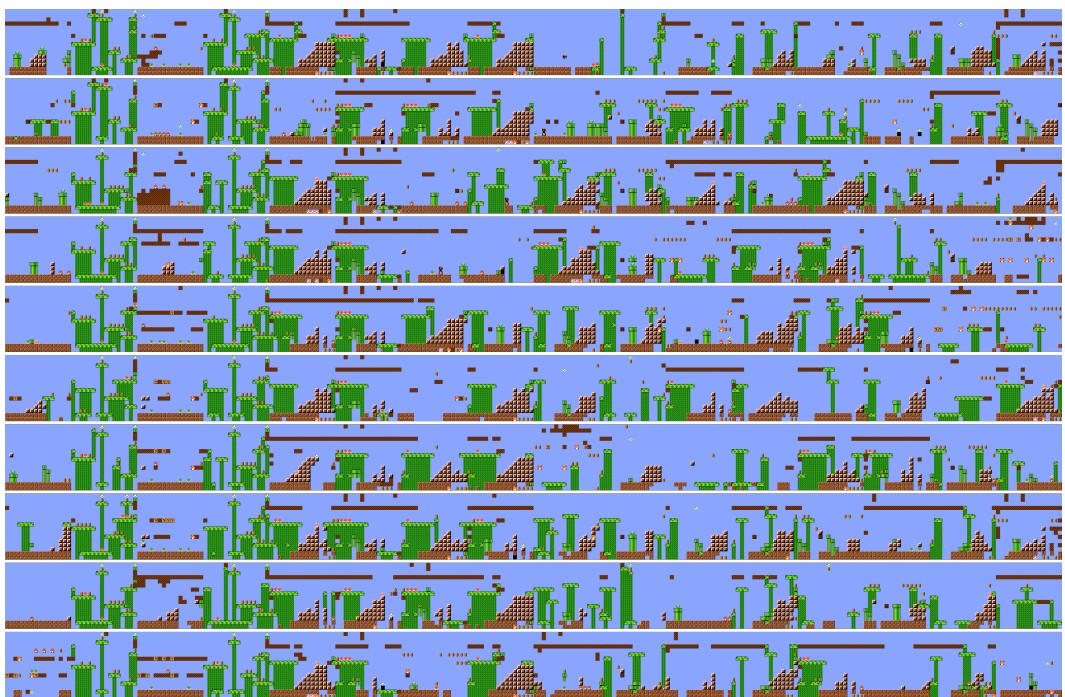

(a) Example levels generated by an NCERL generator trained with $\lambda = 0.5, m = 5$ on the MarioPuzzle task. The reward and diversity scores of this generator are $47.8$ and $2102$, respectively.

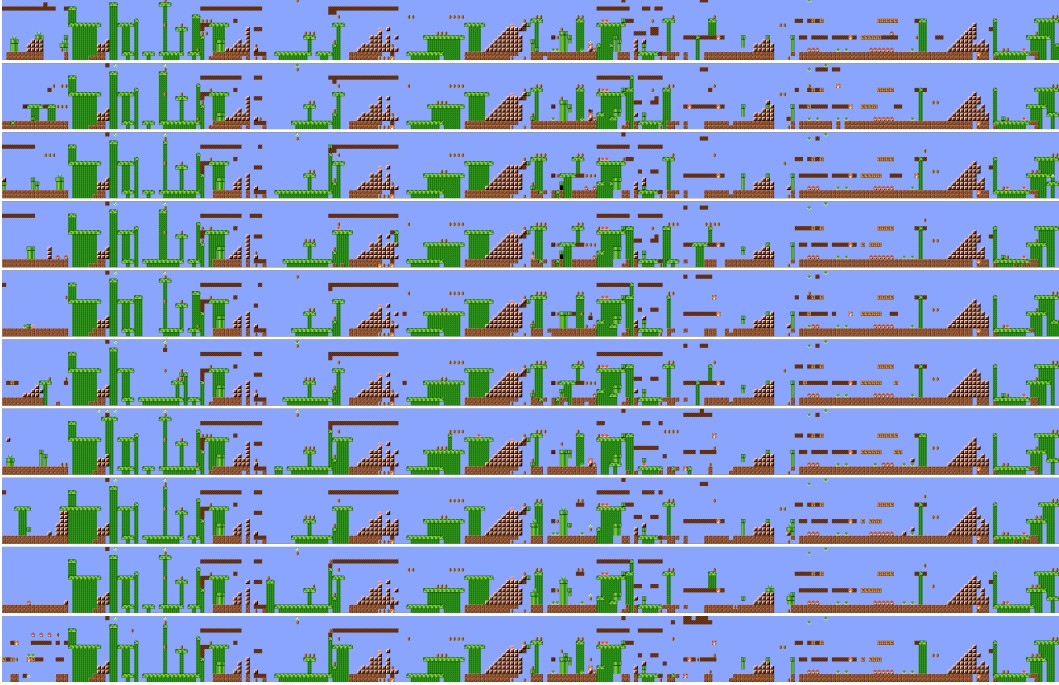

(b) Example levels generated by a SAC generator trained on the MarioPuzzle task. The reward and diversity scores of this generator are $57.4$ and $656.8$, respectively.

Figure 9: Example levels generated by an NCERL generator trained on MarioPuzzle with $\lambda = 0.5, m = 5$ and a SAC generator. SAC generated similar levels while NCERL generated diverse levels.

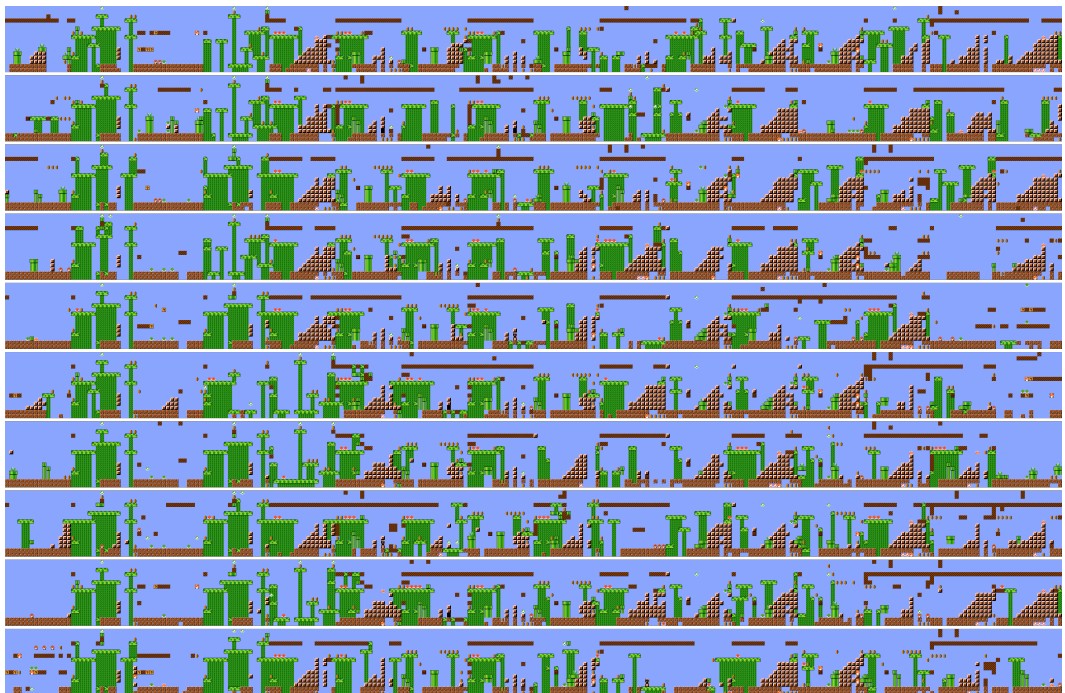

(a) Example levels generated by an NCERL generator trained with $\lambda = 0.3, m = 5$ on the MarioPuzzle task. The reward and diversity scores of this generator are $54.7$ and $1974$, respectively.

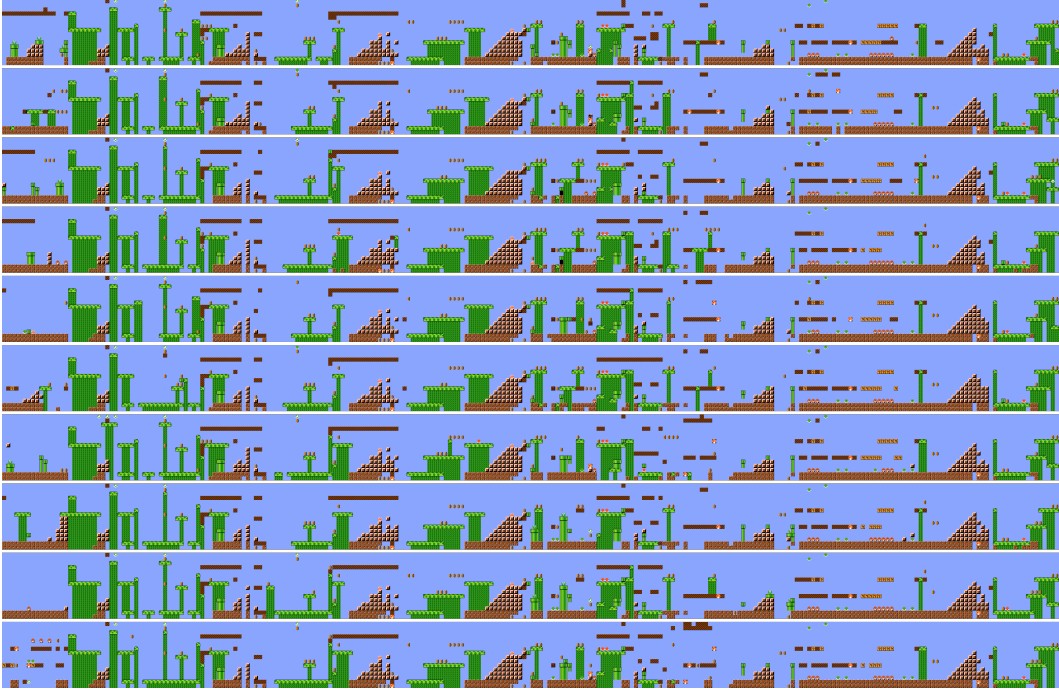

(b) Example levels generated by a SAC generator trained on the MarioPuzzle task. The reward and diversity scores of this generator are $57.4$ and $656.8$, respectively.

Figure 10: Example levels generated by an NCERL generator trained on MarioPuzzle with $\lambda = 0.3, m = 5$ and a SAC generator.

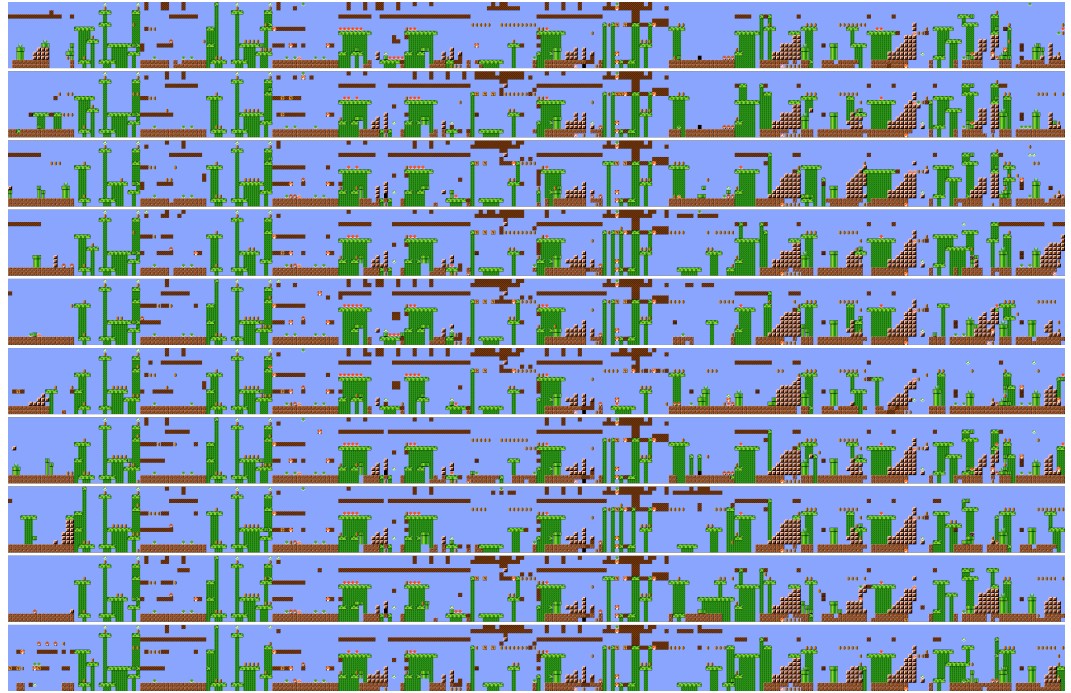

(a) Example levels generated by an NCERL generator trained with $\lambda = 0.1, m = 5$ on the MarioPuzzle task. The reward and diversity scores of this generator are $58.7$ and $1232$, respectively.

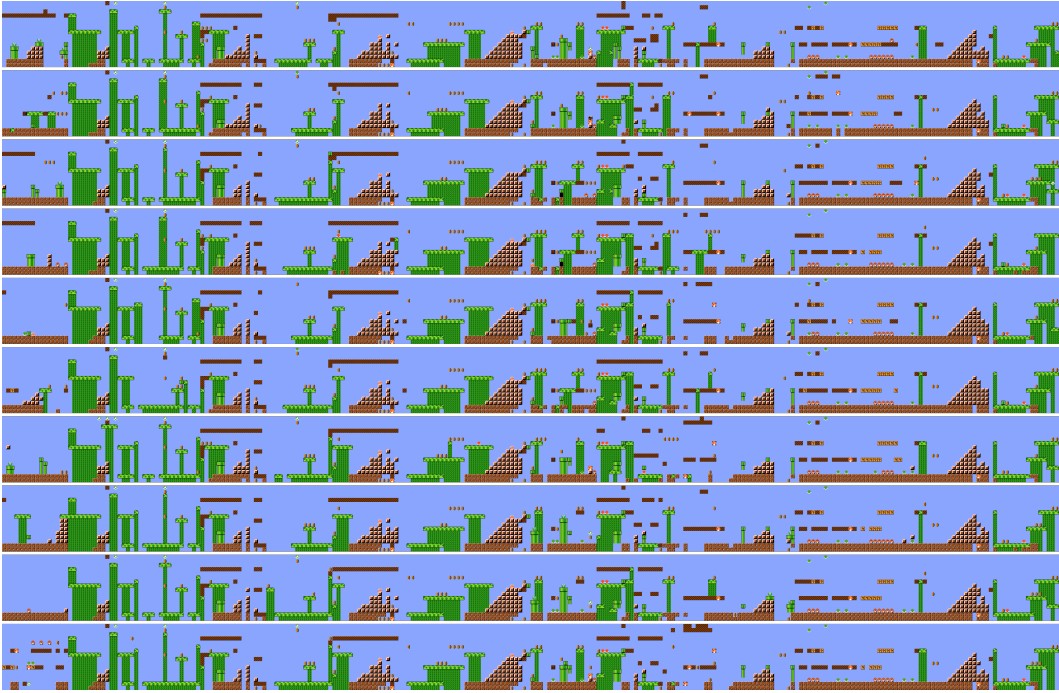

(b) Example levels generated by a SAC generator trained on the MarioPuzzle task. The reward and diversity scores of this generator are $57.4$ and $656.8$, respectively.

Figure 11: Example levels generated by an NCERL generator trained on MarioPuzzle with $\lambda = 0.1, m = 5$ and a SAC generator.

