# OpenReview forum: "Negatively Correlated Ensemble Reinforcement Learning for Online Diverse Game Level Generation"
_ICLR.cc/2024/Conference — ICLR 2024 poster_

### Official Review · Reviewer_cq3K · 2023-10-17

**Soundness:** 3 good
**Presentation:** 2 fair
**Contribution:** 3 good
**Rating:** 6
**Confidence:** 3

**Summary:**

This paper proposes NCERL, an ensemble Reinforcement Learning (RL) method for game level generation with more diversity. NCERL uses a set of different Gaussian actors which outputs actions in the latent space, and the output is decoded into different level segments by a GAN decoder. The final output segment is chosen with the probability generated by a learned selector. To encourage diversity between different actors, a regularizer of Wasserstein distance between policy distributions (and weighted by selector) is added to the reward. The paper also gives a modified gradient and convergence proof on the new reward, as the regularizer depends on the whole policy instead of a single action. On a well known level-generation benchmark, NCERL achieves comparable reward (measuring game-design goals) with better level diversity, and can also perform a well trade-off between diversity and reward by controlling its regularization coefficient.

**Strengths:**

1. As this paper uses a latent space for policy and existing decoder for segment generation, the problem addressed by this paper is not only interesting, but actually quite general: how to achieve a trade-off between policy diversity and performance, with game level generation being one of its real-world applications.

2. The paper has a sound theoretical basis, with a convergence proof for the new reward (which, similar to soft actor-critic, depends on policy distribution instead of action) and a rigorous, modified version of gradient update.

3. The proposed work is scalable, with a well-designed and parallelized reward evaluation framework.

**Weaknesses:**

**Some details of the papers are not presented clearly enough.**

1. Figure 1 has many math symbols and no caption on high-level ideas of each component; in addition, the meaning of $i\leftarrow 2$ in the figure is unclear.

2. There is no clear definition on what a state is in the paper; the readers can only speculate that it is a latent vector from the end of Section 2.1 ("... a randomly sampled latent vector is used as the initial state").

3. There is no description on how Wasserstein distance is calculated. It is true that 2-Wasserstein distance between Gaussian distributions can be easily calculated, but it would be better if the Gaussian property can be emphasized at the beginning of Section 3.2, a formula can be given to make the paper self-contained, and "2-Wasserstein" instead of "Wasserstein" is specified.

4. typos: in conclusion, bettwe -> better.

**Questions:**

I have two questions:

1. In Table 1, the trade-off of NCERL between reward and diversity in some of the environments does not follow the trend; for example, in Mario Puzzle, the diversity with $\lambda$ seems to have two peaks (0.2 and 0.5), and the reward at $\lambda=0.1$ is the worst despite of low diversity. Could the author explain this?

2. Currently, there is only comparison between NCERL, ensemble RL methods and non-ensemble RL methods, and the encoding is over GAN. What is the performance of non-RL solutions, such as scripted (possibly with learnable parameters) solution, or supervised/self-supervised learning over more recent generative models such as diffusion models or VAEs?

---

> ### Author Response · Authors · 2023-11-23
> **Author Response to Reviewer cq3K**
>
> Thank you for the insightful comments and we greatly appreciate your constructive suggestions for improving the presentation. Our paper has been carefully revised according to your suggestions and we also checked through the paper again for typos. The changes are highlighted in blue. For your questions, we hope the following responses address them properly.
>
> **Q1: In Table 1, the trade-off of NCERL between reward and diversity in some of the environments does not follow the trend; for example, in Mario Puzzle, the diversity with λ seems to have two peaks (0.2 and 0.5), and the reward at λ=0.1 is the worst despite of low diversity. Could the author explain this?**
>
> We consider this may be caused by some local optima during the training. Therefore, the training results of NCERL can be a little bit unstable. The black-box decoder further makes the training results hard to predict. The $\lambda$ factor does not affect the trade-off between reward and diversity but also affects the exploration efficiency, making the effect of $\lambda$ on the performance more non-monotonous. The relative poor performance of the generator trained with $\lambda = 0.1$ can be attributed to an abnormal trial which only yielded $30.23$ of average reward and $1723$ of diversity. For future work, we consider integrating our method with multi-objective reinforcement learning [1] to train a set of non-dominated policies with mutable regularisation coefficients. This could potentially bypass the problem of unstable and non-monotonous effects of the regularisation weight. To facilitate a more in-depth analysis of the results, we present the reward and diversity of each independent trial. Full results are presented and discussed in Appendix E.1.
>
> **Q2: Currently, there is only a comparison between NCERL, ensemble RL methods and non-ensemble RL methods, and the encoding is over GAN. What is the performance of non-RL solutions, such as scripted (possibly with learnable parameters) solutions, or supervised/self-supervised learning over more recent generative models such as diffusion models or VAEs?**
>
> Due to the limited time, we are not able to compare our approach with all the non-RL approaches listed by the reviewer, but we made an effort to conduct a meaningful comparison. We use the code of [2] to train diffusion models (DDPM) for five independent trials. Specifically, we utilized the code from [2] to train DDPM for five independent trials. In this process, we randomly sampled 25 noises to generate 25 level segments, then concatenated them with an initial segment used in the RL generator testing. This procedure was applied to each of the 500 initial segments used for testing RL generators, resulting in a test set of 500 levels. We evaluated these test sets in terms of both reward and diversity.
>
> |Task|Criterion|DDPM|$\lambda$=$0.0$|$\lambda$=$0.1$|$\lambda$=$0.2$|$\lambda$=$0.3$|$\lambda$=$0.4$|$\lambda$=$0.5$|
> |-------------|-----------|-----------------|-----------------|-----------------|-----------------|-----------------|-----------------|-----------------|
> |MarioPuzzle|Reward|*-29.45*|**55.24**|51.42|53.78|53.22|54.59|53.26|
> |MarioPuzzle|Diversity|1630|*1342*|1570|1940|1688|1698|**1967**|
> |MultiFacet|Reward|*-119.4*|**46.39**|46.16|45.35|37.87|40.86|35.77|
> |MultiFacet|Diversity|**1630**|*401.6*|492.3|620.2|1024|889.6|1142|
>
> According to the results, DDPM exhibited good diversity scores but performed poorly in terms of reward. This is attributed to the fact that the training of DDPM does not consider reward functions that evaluate the quality of generated levels. To our knowledge, DDPM can not optimise certain objectives to cater to customised objectives, while the scripted method relies on domain knowledge and needs significant development costs. The generated samples of DDPM are available in our anonymous code repository (https://anonymous.4open.science/r/NCERL-Diverse-PCG-4F25/, the generation_results folder).
>
> [1] Hayes, Conor F., et al. "A practical guide to multi-objective reinforcement learning and planning." Autonomous Agents and Multi-Agent Systems 36.1 (2022): 26.
>
> [2] Lee, Hyeon Joon, and Edgar Simo-Serra. "Using Unconditional Diffusion Models in Level Generation for Super Mario Bros." 2023 18th International Conference on Machine Vision and Applications. IEEE, 2023.

---

> > ### Author Response · Authors · 2023-11-23
> > **Follow up response to other comments**
> >
> > For your suggestions about the presentation, we have addressed them carefully in the revision.  Our changes in the paper are highlighted in blue.
> >
> > **1.Figure 1 has many math symbols and no caption on high-level ideas of each component; in addition, the meaning of $i \gets 2$ in the figure is unclear.**
> >
> > We have updated Figure 1 and its caption to explain the components and math symbols.
> >
> > **2.There is no clear definition on what a state is in the paper; the readers can only speculate that it is a latent vector from the end of Section 2.1 ("... a randomly sampled latent vector is used as the initial state").**
> >
> > We are sorry for such unclarity in the paper.  We have revised the description of the domain (state space, action space, reward criterion and diversity criterion) to make it clearer. In the revised paper, Section 3 “Problem Formulation” describes state and action. A state is a concatenated vector of a fixed number of latent vectors from recently generated segments. If there are not enough segments ($< n$) to construct a complete state, zeros will be padded in the vacant entries.
> >
> > **3.There is no description on how Wasserstein distance is calculated. It is true that 2-Wasserstein distance between Gaussian distributions can be easily calculated, but it would be better if the Gaussian property can be emphasized at the beginning of Section 3.2, a formula can be given to make the paper self-contained, and "2-Wasserstein" instead of "Wasserstein" is specified.**
> >
> > We have updated Section 3.2 as suggested by the reviewer. In the revision, Appendix C.2 is added to provide the formulation of 2-Wasserstein distance and how it can be calculated between Gaussian distributions.
> >
> > **4.typos: in conclusion, bettwe -> better.**
> >
> > We have fixed it and checked through the paper for typos.

---

### Official Review · Reviewer_t3p4 · 2023-10-19

**Soundness:** 3 good
**Presentation:** 3 good
**Contribution:** 3 good
**Rating:** 6
**Confidence:** 4

**Summary:**

The paper proposes an ensemble reinforcement learning approach for generating diverse game levels. The approach uses multiple sub-policies to generate different alternative level segments, and stochastically selects one of them following a selector model. The paper also integrates a novel policy regularisation technique, which is a negative correlation regularisation that increases the distances between the decision distributions determined by each pair of actors. The regularisation is optimised using regularised versions of the policy iteration and policy gradient, which provide general methodologies for optimising policy regularisation in a Markov decision process. The paper's contributions are:
1. The proposed ensemble reinforcement learning approach for generating diverse game levels.
2. The novel policy regularisation technique that encourages the sub-policies to explore different regions of the state-action space.
3. The regularised versions of the policy iteration and policy gradient algorithms that provide general methodologies for optimising policy regularisation in a Markov decision process.

**Strengths:**

originality: the paper proposes a novel approach for generating diverse game levels using ensemble reinforcement learning and policy regularisation. The paper develops two theorems to provide general methodologies for optimizing policy regularisation in a Markov decision process. The first theorem is a regularised version of the policy iteration algorithm, which is a classic algorithm for solving MDPs. The second theorem is a regularised version of the policy gradient algorithm, which is another classic algorithm for solving MDPs.

quality: the paper provides a detailed description of the proposed approach and the regularisation technique. The paper also provides theoretical proofs of the regularised versions of the policy iteration and policy gradient algorithms.

clarity: the paper is well-written and easy to follow. The authors provide clear explanations of the proposed approach and the regularisation technique.

**Weaknesses:**

the proposed approach assumes that the reward function is known and fixed. However, in practice, the reward function may be unknown or may change over time. Therefore, the proposed approach may not be applicable in such scenarios.

the paper only considers a single game genre (platformer) and a single game engine (Super Mario Bros.). The proposed approach may not be directly applicable to other game genres or engines.

**Questions:**

Q1: can the proposed approach be directly applicable to 3D game levels or other types of game content? or what are the difficulties in this extension, such as complex reward design or high computational burden?

---

> ### Author Response · Authors · 2023-11-23
> **Author Response to Reviewer t3p4**
>
> Thank you for providing insightful comments. We hope the following response addresses your concerns. Our paper is revised and updated according to reviewers’ comments, the changes are highlighted in blue.
>
> **Q1: can the proposed approach be directly applicable to 3D game levels or other types of game content? or what are the difficulties in this extension, such as complex reward design or high computational burden?**
>
> Our method can be directly applicable to 3D game levels like Minecraft generation [1] and other types of game content like music [2] and narrative generation [3] since RL has been applied to generate that content. Those applications could benefit from our proposed method since we only change the RL algorithm while maintaining the problem setting unchanged.
>
> When applying our approach to other scenarios, the design of the reward can be challenging, as indicated by the reviewer. However, the procedural content generation (PCG) community has proposed a range of evaluation metrics [4], which can serve as the reward function in our framework. Taking Minecraft as an example again, Jiang et al. [1] have proposed some reward functions to train controllable RL-based game level generators, while there is also a set of evaluation metrics verified based on human evaluation scores [5], those metrics could be used as reward functions. For the issue of high computational burden, we have devised and implemented an asynchronous framework to speed up the training. On the other hand, the use of an action decoder, proposed in [6], contributes to faster generation speeds. There are some generative models that can generate high-quality 3D game levels, such as world-GAN [7] for Minecraft world generation and VAE-GAN for 3D indoor scene generation [8]. By employing these models as decoders, our method can be directly applied to the corresponding application scenarios with rapid generation speeds.
>
> [1] Jiang, Zehua, et al. "Learning Controllable 3D Level Generators." Proceedings of the 17th International Conference on the Foundations of Digital Games. 2022.
>
> [2] Jaques, Natasha, et al. "Generating music by fine-tuning recurrent neural networks with reinforcement learning." (2016).
>
> [3] Huang, Qiuyuan, et al. "Hierarchically structured reinforcement learning for topically coherent visual story generation." Proceedings of the AAAI Conference on Artificial Intelligence. Vol. 33. No. 01. 2019.
>
> [4] Shaker, Noor, Julian Togelius, and Mark J. Nelson. "Procedural content generation in games." (2016): 978-3.
>
> [5] Hervé, Jean-Baptiste, and Christoph Salge. "Comparing PCG metrics with Human Evaluation in Minecraft Settlement Generation." Proceedings of the 16th International Conference on the Foundations of Digital Games. 2021.
>
> [6] Shu, Tianye, Jialin Liu, and Georgios N. Yannakakis. "Experience-driven PCG via reinforcement learning: A Super Mario Bros study." 2021 IEEE Conference on Games (CoG). IEEE, 2021.
>
> [7] Awiszus, Maren, Frederik Schubert, and Bodo Rosenhahn. "World-gan: a generative model for minecraft worlds." 2021 IEEE Conference on Games (CoG). IEEE, 2021.
>
> [8] Li, Shuai, and Hongjun Li. "Deep Generative Modeling Based on VAE-GAN for 3D Indoor Scene Synthesis." International Journal of Computer Games Technology 2023 (2023).

---

> > ### Author Response · Authors · 2023-11-23
> > **Follow up response to other comments**
> >
> > Besides the above response to the reviewer's question, we would like to clarify the significance of this work regarding the reviewer's comments.
> >
> > Regarding the comment "**the proposed approach assumes that the reward function is known and fixed. However, in practice, the reward function may be unknown or may change over time. Therefore, the proposed approach may not be applicable in such scenarios.**": Although the reward function used in this paper is known and fixed, such a problem setting has extensive application scenarios in the domain of game content generation [1,2,3,6]. However, the reviewer has raised an interesting future research direction. Our proposed method can be made compatible with other approaches to tackle the challenge of unknown or unfixed rewards. For the issue of unknown rewards, inverse reinforcement learning can be used to build some reward functions from human demonstrations [9]. In the context of game content generation, it is possible to collect human-authored levels as demonstrations and learn a reward model from them with inverse reinforcement learning. Our method can be combined with inverse reinforcement learning to address tasks with unknown rewards. When facing the challenge of unfixed rewards, integrating reward estimation and replay sampling techniques [10] into our framework can be a viable solution. By combining our approach with these techniques, one can handle game content generation tasks with unknown or unfixed rewards.
> >
> > Regarding the comment "**the paper only considers a single game genre (platformer) and a single game engine (Super Mario Bros.). The proposed approach may not be directly applicable to other game genres or engines.**": This paper employs Super Mario Bros. as the benchmark for our approach, as it is commonly used and representative within the Procedural Content Generation (PCG) community, and it is open-source. For more details on the potential applications of our approach to other games, please refer to our response to Q1.
> >
> > [9] Arora, Saurabh, and Prashant Doshi. "A survey of inverse reinforcement learning: Challenges, methods and progress." Artificial Intelligence 297 (2021): 103500.
> >
> > [10] Chen, Shi-Yong, et al. "Stabilizing reinforcement learning in dynamic environment with application to online recommendation." Proceedings of the 24th ACM SIGKDD International Conference on Knowledge Discovery & Data Mining. 2018.

---

### Official Review · Reviewer_qX6Q · 2023-10-31

**Soundness:** 3 good
**Presentation:** 3 good
**Contribution:** 2 fair
**Rating:** 5
**Confidence:** 3

**Summary:**

**Problem Setting**

We want to do level generation but we want to induce some diversity in how the levels are generated. Here the policies define level generators which build the level through an MDP.

**Algorithm / NN Structure**

The policies are defined as mixtures of Gaussians, implemented by creating a set of sub-policies along with a weighting. The weighting itself id modelled by a selector policy, creating a form of hierarchy.

The sub-policies are regularized to be diverse from each other using a Wasserstein distance. This distance is clipped, encouraging policies to be diverse only if their decisions are too close. Sub-policies have Gaussian action heads. Regularization is implemented as an auxilliary reward.

A regularized version of policy iteration of the policy gradient are presented. Thus the agent is trained not only to optimize for diversity in the current timestep, but in future timesteps via a reguarlized value function.

The practical implementation is built off SAC.

Experiments are presented on the Mario level generation benchmark. Results show that NCERL is able to achieve comparable reward to other methods, and outperform in terms of diversity.

**Strengths:**

This paper proposes a clean and thorough study of a method to induce diverse level generation. The idea is to define policies as a mixture of sub-policies, then regularize those sub-policies so that diversity is increased. While this is a straightforward idea, it is especially applicable in a domain such as level generation where diversity is desired in itself rather than as simply a means towards exploration. The quality of the writing and presentation is solid and clear. Theoretical results are presented re-deriving the policy iteration and policy gradient update explicitly in terms of regularizing the diversity between sub-policies, and proofs are presented regarding convergence. The significance of this work stems from its thorough theoretical contributions.

**Weaknesses:**

Because the experiments are largely domain-specific and improve on diversity rather than pure performance, the significance of this work is limited.

While there are novel derivations and a clean interpretation of regularizing the policy gradient, the idea of representing an agent as sub-policies has been explored in fields such as skill discovery and hierarchical reinforcement learning, which were not referenced in this work.

The description of the domain is unclear to me as a reader, e.g. what is the action space of an agent generating Mario levels? What are the criteria used to evaluate reward and diversity? I would have liked to see examples of the generated levels.

**Questions:**

See above for questions related to the experimental section.

The section on asynchronous evaluation seems orthogonal to the main contribution of the work. Asynchronous RL has been explored in the actor-critic setting (e.g. A3C), which this work uses as it builds off SAC. Is there a specific connection between the asynchronous implementation and the novel contribution here?

What does the behavior of the weighting-selector policy look like? It would provide more clarity into the method to showcase how often this selector policy utilizes specific sub-policies, or if certain sub-policies go unused.

It may help to label the other comparison methods in Figure 4.

---

> ### Author Response · Authors · 2023-11-23
> **Author Response to Reviewer qX6Q**
>
> Thank you for providing insightful comments. We are especially encouraged by your appreciation of our theoretical contribution. We hope the following response addresses your concerns. Our paper is revised and updated according to reviewers' comments, the changes are highlighted in blue.
>
> **Q1: The section on asynchronous evaluation seems orthogonal to the main contribution of the work. Asynchronous RL has been explored in the actor-critic setting (e.g. A3C), which this work uses as it builds off SAC. Is there a specific connection between the asynchronous implementation and the novel contribution here?**
>
> Indeed, the asynchronous evaluation is orthogonal to the main contribution of this work. Therefore, we removed it from the contribution statements in Section 1 of the revised paper.
>
> **Q2: What does the behaviour of the weighting-selector policy look like? It would provide more clarity into the method to showcase how often this selector policy utilizes specific sub-policies, or if certain sub-policies go unuse**
>
> If the $\lambda$ used in training is large, the probabilities of selecting sub-policies change over time step, and all the sub-policies are used. While if the $\lambda$ is small, it is possible that some sub-policies go unused. To showcase the selection probabilities, we pick two NCERL generators trained with $m=5, \lambda= 0.5$ and $m = 5, \lambda=0.1$ and record their selection probabilities during stochastically generating two levels from the same initial state, respectively.
>
> Regarding the generator trained with $\lambda = 0.5$, all the sub-policies are activated multiple times within the two trials. Regarding the generator trained with $\lambda = 0.1$, some of the selection probability is near zero and sub-policies 2 and 3 are never used within the two trials. As the regularisation coefficient $\lambda$ increases, the probabilities of selecting sub-policies become more uniform. The tables are included in Appendix E.2  of our revised paper with the images of generated levels in those two trials. We also present them as follows for reference.

---

> > ### Author Response · Authors · 2023-11-23
> > **Follow up (tables)**
> >
> > **Selection probabilities of an NCERL generator trained with $\lambda=0.5$ and $m=5$, trial 1, bold texts indicate their corresponding sub-policies are activated at their corresponding time steps.**
> >
> > ||$t=1$|$t=2$|$t=3$|$t=4$|$t=5$|$t=6$|$t=7$|$t=8$|$t=9$|$t=10$|$t=11$|$t=12$|$t=13$|$t=14$|$t=15$|
> > |-|-|-|-|-|-|-|-|-|-|-|-|-|-|-|-|
> > |$\beta_1$|$\approx0$|$.062$|$.084$|$.138$|$.094$|$.147$|$.147$|$.144$|$.171$|$.137$|$.179$|$.184$|$\boldsymbol{.208}$|$.206$|$.209$|
> > |$\beta_2$|$.357$|$\boldsymbol{.260}$|$.242$|$.218$|$\boldsymbol{.238}$|$.218$|$.222$|$.225$|$.205$|$.241$|$.209$|$\boldsymbol{.209}$|$.188$|$.189$|$.190$|
> > |$\beta_3$|$\boldsymbol{.239}$|$.216$|$.225$|$.233$|$.247$|$.222$|$.233$|$\boldsymbol{.236}$|$.211$|$\boldsymbol{.191}$|$.200$|$.208$|$.195$|$.211$|$.195$|
> > |$\beta_4$|$.404$|$.259$|$\boldsymbol{.254}$|$.196$|$.237$|$.219$|$\boldsymbol{.205}$|$.203$|$\boldsymbol{.201}$|$.259$|$.211$|$.188$|$.196$|$\boldsymbol{.178}$|$\boldsymbol{.197}$|
> > |$\beta_5$|$\approx0$|$.202$|$.195$|$\boldsymbol{.216}$|$.184$|$\boldsymbol{.193}$|$.194$|$.192$|$.212$|$.172$|$\boldsymbol{.201}$|$.211$|$.212$|$.216$|$.208$|
> >
> > **Selection probabilities of an NCERL generator trained with $\lambda=0.5$ and $m=5$, trial 2, bold texts indicate their corresponding sub-policies are activated at their corresponding time steps.**
> >
> > ||$t=1$|$t=2$|$t=3$|$t=4$|$t=5$|$t=6$|$t=7$|$t=8$|$t=9$|$t=10$|$t=11$|$t=12$|$t=13$|$t=14$|$t=15$|
> > |-|-|-|-|-|-|-|-|-|-|----|---------|-------|---|----|----|
> > |$\beta_1$|$\approx0$|$.035$|$.074$|$.159$|$.132$|$.149$|$.149$|$.140$|$.164$|$\boldsymbol{.169}$|$.178$|$\boldsymbol{.198}$|$\boldsymbol{.155}$|$.120$|$.203$|
> > |$\beta_2$|$\boldsymbol{.357}$|$\boldsymbol{.272}$|$.264$|$.212$|$.233$|$.219$|$.218$|$.226$|$.214$|$.203$|$.201$|$.192$|$.213$|$.259$|$\boldsymbol{.191}$|
> > |$\beta_3$|$.239$|$.192$|$.214$|$.218$|$.241$|$.228$|$.222$|$.231$|$.195$|$.212$|$.218$|$.200$|$.218$|$.218$|$.198$|
> > |$\beta_4$|$.404$|$.285$|$.276$|$\boldsymbol{.210}$|$.202$|$.209$|$.208$|$\boldsymbol{.204}$|$\boldsymbol{.227}$|$.209$|$.191$|$.199$|$.216$|$\boldsymbol{.278}$|$.198$|
> > |$\beta_5$|$\approx0$|$.216$|$\boldsymbol{.173}$|$.202$|$\boldsymbol{.193}$|$\boldsymbol{.196}$|$\boldsymbol{.203}$|$.199$|$.199$|$.207$|$\boldsymbol{.213}$|$.211$|$.198$|$.125$|$.210$|
> >
> >
> > **Selection probabilities of an NCERL generator trained with $\lambda=0.1$ and $m=5$, trial 1, bold texts indicate their corresponding sub-policies are activated at their corresponding time steps.**
> >
> > ||$t=1$|$t=2$|$t=3$|$t=4$|$t=5$|$t=6$|$t=7$|$t=8$|$t=9$|$t=10$|$t=11$|$t=12$|$t=13$|$t=14$|$t=15$|
> > |--|--|--|--|-|-|---|----|--|--|-|---|--|--|-|-|
> > |$\beta_1$|$\boldsymbol{1.00}$|$.382$|$\boldsymbol{.531}$|$.501$|$\boldsymbol{.609}$|$.522$|$.483$|$\boldsymbol{.539}$|$.560$|$\boldsymbol{.513}$|$\boldsymbol{.610}$|$.711$|$\boldsymbol{.661}$|$\boldsymbol{.687}$|$\boldsymbol{.522}$|
> > |$\beta_2$|$\approx0$|$\approx0$|$\approx0$|$\approx0$|$\approx0$|$\approx0$|$\approx0$|$\approx0$|$\approx0$|$\approx0$|$\approx0$|$\approx0$|$\approx0$|$\approx0$|$.009$|
> > |$\beta_3$|$\approx0$|$\approx0$|$\approx0$|$\approx0$|$\approx0$|$\approx0$|$\approx0$|$\approx0$|$.003$|$\approx0$|$.016$|$.005$|$.003$|$\approx0$|$.035$|
> > |$\beta_4$|$\approx0$|$\boldsymbol{.618}$|$.469$|$\boldsymbol{.499}$|$.391$|$\boldsymbol{.478}$|$\boldsymbol{.517}$|$.461$|$\boldsymbol{.437}$|$.487$|$.370$|$\boldsymbol{.284}$|$.336$|$.313$|$.420$|
> > |$\beta_5$|$\approx0$|$\approx0$|$\approx0$|$\approx0$|$\approx0$|$\approx0$|$\approx0$|$\approx0$|$\approx0$|$\approx0$|$.003$|$\approx0$|$\approx0$|$\approx0$|$.015$|
> >
> >
> > **Selection probabilities of an NCERL generator trained with $\lambda=0.1$ and $m=5$, trial 2, bold texts indicate their corresponding sub-policies are activated at their corresponding time steps.**
> >
> > ||$t=1$|$t=2$|$t=3$|$t=4$|$t=5$|$t=6$|$t=7$|$t=8$|$t=9$|$t=10$|$t=11$|$t=12$|$t=13$|$t=14$|$t=15$|
> > |-------|--|----|-|----|----|--|-|------|-----|--------|-----|----------|----------|-----------|---------|
> > |$\beta_1$|$\boldsymbol{1.00}$|$.435$|$.529$|$.502$|$\boldsymbol{.502}$|$\boldsymbol{.450}$|$.430$|$\boldsymbol{.534}$|$\boldsymbol{.502}$|$\boldsymbol{.470}$|$.453$|$.350$|$\boldsymbol{.506}$|$\boldsymbol{.474}$|$.400$|
> > |$\beta_2$|$\approx0$|$\approx0$|$\approx0$|$\approx0$|$\approx0$|$\approx0$|$\approx0$|$\approx0$|$\approx0$|$\approx0$|$.012$|$\approx0$|$\approx0$|$\approx0$|$.012$|
> > |$\beta_3$|$\approx0$|$\approx0$|$\approx0$|$\approx0$|$\approx0$|$\approx0$|$\approx0$|$\approx0$|$\approx0$|$\approx0$|$.046$|$\approx0$|$.013$|$\approx0$|$.026$|
> > |$\beta_4$|$\approx0$|$\boldsymbol{.565}$|$\boldsymbol{.471}$|$\boldsymbol{.498}$|$.498$|$.550$|$\boldsymbol{.570}$|$.466$|$.498$|$.530$|$\boldsymbol{.468}$|$\boldsymbol{.650}$|$.479$|$.526$|$.546$|
> > |$\beta_5$|$\approx0$|$\approx0$|$\approx0$|$\approx0$|$\approx0$|$\approx0$|$\approx0$|$\approx0$|$\approx0$|$\approx0$|$.021$|$\approx0$|$.002$|$\approx0$|$\boldsymbol{.016}$|

---

> > > ### Author Response · Authors · 2023-11-23
> > > **Follow up response to the questions**
> > >
> > > **Q3: It may help to label the other comparison methods in Figure 4.**
> > >
> > > Thank you for your constructive suggestion. We have added the legends to label the algorithms and $\lambda$ values in Figure 4.
> > >
> > > **Q4: The description of the domain is unclear to me as a reader, e.g. what is the action space of an agent generating Mario levels? What are the criteria used to evaluate reward and diversity?**
> > >
> > > We have revised the description of the domain (state space, action space, reward criterion and diversity criterion) to make it clearer. In the revised paper, Section 3 “Problem Formulation” describes state and action, Section 6.1 "Online Level Generation Tasks" briefly describes the reward and Section 6.2 "Performance Criteria" introduces the reward and diversity criteria. Due to the page limit, more details including the formulations of reward function and performance criteria are presented in Appendix D.4 and D.1. The state space, action space, reward criterion, and diversity criterion are described as follows.
> > >
> > > **State space**:  A $dn$ dimensional continuous space, where $d$ is the dimensionality of the latent vector of the action decoder and $n$ is the number of recently generated segments being considered in the reward function. A state is a concatenated vector of a fixed number of latent vectors of recently generated segments. If there are not enough segments have been generated (< n) to construct a state, zeros will be padded in the vacant entries.
> > >
> > > **Action space**: A $d$-dimensional continuous space. An action is a latent vector which can be decoded into a level segment by the decoder. The decoder is a trained GAN in this work.
> > >
> > > **Reward criterion**: reward criterion for a generator is calculated as $R = \sum_{t=1}^h R_t$ for each level, i.e., each MDP trajectory, then averaged over all levels generated for testing performance. The reward functions are adopted from previous level generation papers. We described them in Section 6.1 "Online Level Generation Tasks" formulated and described them in Appendix D.1. The description has been revised to make it clearer.
> > >
> > > **Diversity criterion**: Diversity score of a generator is calculated as $D = \frac{2}{n(n-1)} \sum_{i=1}^n \sum_{j \neq i} \Delta(x_i, x_j)$, where $\Delta(\cdot, \cdot)$ indicates the Hamming distance, i.e., how many different tiles are there between the two levels to be compared; and $x_i$, $x_j$ indicate the $i^{\text{th}}$ one and the $j^{\text{th}}$ one in the $n$ levels generated for testing performance.
> > >
> > > **Q5: I would have liked to see examples of the generated levels.**
> > >
> > > We have added some of the generated levels in Appendix E.3 of our revised paper. On both tasks, the levels generated by NCERL generators trained with $\lambda=0.1$, $\lambda=0.3$, and $\lambda=0.5$ are visualised and compared to the ones generated by SAC. Those examples show SAC generate similar levels while our proposed NCERL generate diverse levels. Complete generation samples of all trained generators are uploaded to our anonymous code repository (https://anonymous.4open.science/r/NCERL-Diverse-PCG-4F25/, the generation_results folder).

---

> > > > ### Author Response · Authors · 2023-11-23
> > > > **Follow up response to other comments**
> > > >
> > > > Besides the above point-to-point responses to the reviewer's questions, we would like to clarify the significance of this work regarding the reviewer's comments.
> > > >
> > > > Regarding the comment "**Because the experiments are largely domain-specific and improve on diversity rather than pure performance, the significance of this work is limited.**": Although our work is domain-specific, online level generation is a broad and crucial area with extensive industrial applications. Procedural content generation via reinforcement learning is actively researched and applied in various gaming domains, including 2D [1] and 3D games [2], as well as virtual reality games [3]. On the other hand, many commercial games feature online level generation, for example, Minecraft, No Man’s Sky and a wide range of roguelike games like Spelunky and Hades. While traditional methods typically rely on human-crafted rules, machine learning-based methods can largely reduce the development cost [4-7]. Moreover, though we do not improve on reward, diversity itself is a goal of interest in not only game content generation, but also other RL directions like multi-agent RL [8] and quality-diversity RL [9].
> > > >
> > > > [1] Khalifa, Ahmed, et al. "PCGRL: Procedural content generation via reinforcement learning." Proceedings of the AAAI Conference on Artificial Intelligence and Interactive Digital Entertainment. Vol. 16. No. 1. 2020.
> > > >
> > > > [2] Jiang, Zehua, et al. "Learning Controllable 3D Level Generators." Proceedings of the 17th International Conference on the Foundations of Digital Games. 2022.
> > > >
> > > > [3] Mahmoudi-Nejad, Athar, Matthew Guzdial, and Pierre Boulanger. "Arachnophobia exposure therapy using experience-driven procedural content generation via reinforcement learning (EDPCGRL)." Proceedings of the AAAI Conference on Artificial Intelligence and Interactive Digital Entertainment. Vol. 17. No. 1. 2021.
> > > >
> > > > [4] Yannakakis, Georgios N., and Julian Togelius. Artificial intelligence and games. Vol. 2. New York: Springer, 2018.
> > > >
> > > > [5] Shaker, Noor, Julian Togelius, and Mark J. Nelson. "Procedural content generation in games." (2016): 978-3.
> > > >
> > > > [6] Liu, Jialin, et al. "Deep learning for procedural content generation." Neural Computing and Applications 33.1 (2021): 19-37.
> > > >
> > > > [7] Guzdial, Matthew, Sam Snodgrass, and Adam J. Summerville. Procedural Content Generation Via Machine Learning: An Overview. Springer, 2022.
> > > >
> > > > [8] Cui, B., Lupu, A., Sokota, S., Hu, H., Wu, D. J., & Foerster, J. N. (2022, September). Adversarial Diversity in Hanabi. In The Eleventh International Conference on Learning Representations.
> > > >
> > > > [9] Wu, Shuang, et al. "Quality-Similar Diversity via Population Based Reinforcement Learning." The Eleventh International Conference on Learning Representations. 2022.
> > > >
> > > > Regarding the comment "**While there are novel derivations and a clean interpretation of regularizing the policy gradient, the idea of representing an agent as sub-policies has been explored in fields such as skill discovery and hierarchical reinforcement learning, which were not referenced in this work.**": Indeed, the idea of representing an agent as sub-policies has been explored in skill discovery and hierarchical reinforcement learning. We have added the relevant reference [10,11]  in the "Population-based RL" part of Section 2 of our revised paper.
> > > >
> > > > [10] Pateria, Shubham, et al. "Hierarchical reinforcement learning: A comprehensive survey." ACM Computing Surveys (CSUR) 54.5 (2021): 1-35.
> > > >
> > > > [11] Konidaris, George, and Andrew Barto. "Skill discovery in continuous reinforcement learning domains using skill chaining." Advances in neural information processing systems 22 (2009).

---

### Official Review · Reviewer_NGar · 2023-11-01

**Soundness:** 3 good
**Presentation:** 3 good
**Contribution:** 3 good
**Rating:** 6
**Confidence:** 3

**Summary:**

This paper introduces a method for online generating diverse game levels through an ensemble of negatively correlated RL generators. The authors derived a policy update operator under the diversity bonus. Apart from that, the authors propose an async framework for speeding up the training. Experiments show that the method is able to generate a wide range of policies through tunning the diversity coefficient $\lambda$.

**Strengths:**

1. The paper is written in clarity. Hypotheses are well supported by the experiments.
2. Originality looks good to me (or maybe I am not following the OLG line of research, but I study MARL diversity, in which no noticeable significantly similar methods to my knowledge)

**Weaknesses:**

To my understanding, this method adds a reward bonus/diversity constraint to the diversity among the policies, where the proof is kind of established in the literature. The effect of adding diversity regularization is similar to the quality diversity methods, where you are pursuing optimal in the new reward space. The role of $\lambda$ is close to the Lagrange multiplier in the dual formulation of the original problem with a diversity constraint. I think it is ok to include them as contributions to the paper but building theoretical analysis on the interactions among ensemble policies or regularization effect would be more interesting. My ratings are subject to change.

**Questions:**

It is similar to (adversarial) diversity in populations of policies that learn incompatible policies or impose distance(entropy) regularization. Can the authors provide their view of how it compares to population-based methods with diversity regularization?

[1] Xing, D., Liu, Q., Zheng, Q., Pan, G., & Zhou, Z. H. (2021). Learning with Generated Teammates to Achieve Type-Free Ad-Hoc Teamwork. In IJCAI (pp. 472-478).

[2] Lupu, A., Cui, B., Hu, H. &amp; Foerster, J.. (2021). Trajectory Diversity for Zero-Shot Coordination. <i>Proceedings of the 38th International Conference on Machine Learning</i>, in <i>Proceedings of Machine Learning Research</i> 139:7204-7213 Available from https://proceedings.mlr.press/v139/lupu21a.html.

[3] Cui, B., Lupu, A., Sokota, S., Hu, H., Wu, D. J., & Foerster, J. N. (2022, September). Adversarial Diversity in Hanabi. In The Eleventh International Conference on Learning Representations.

[4] Rahman, A., Fosong, E., Carlucho, I., & Albrecht, S. V. (2023). Generating Teammates for Training Robust Ad Hoc Teamwork Agents via Best-Response Diversity. Transactions on Machine Learning Research.

[5] Charakorn, R., Manoonpong, P., & Dilokthanakul, N. (2022, September). Generating Diverse Cooperative Agents by Learning Incompatible Policies. In The Eleventh International Conference on Learning Representations.

[6] Rahman, A., Cui, J., & Stone, P. (2023). Minimum Coverage Sets for Training Robust Ad Hoc Teamwork Agents. arXiv preprint arXiv:2308.09595.

---

> ### Author Response · Authors · 2023-11-23
> **Author Response to Reviewer NGar**
>
> Thank you for providing insightful comments. We hope the following response addresses your concerns. Our paper has been revised according to reviewers' comments, the changes are highlighted in blue.
>
> **Q1: It is similar to (adversarial) diversity in populations of policies that learn incompatible policies or impose distance(entropy) regularization. Can the authors provide their view of how it compares to population-based methods with diversity regularization?**
>
> We appreciate the recommended papers from the reviewers; they have been very helpful. We have checked all of them and found [2,3,5] are the most related ones and have included them in the "Population-based RL" (Renamed from "Policy Ensemble in RL") part of Section 2 in our revised paper. This section also includes a discussion of population diversity in ensemble RL. For papers [2], [3] and [5], we discuss them as follows.
>
> * Paper [2] uses JS divergence of individual agents’ trajectories as a diversity regularisation. It is similar to our diversity regularisation, while we use the regularisation with a different distance metric and to the decision distribution rather than trajectories.
>
> * Paper [3] considers an adversarial diversity, which makes a policy different from an “repulser” policy. This is realised by modifying the TD target, while our approach uses distance between decision distributions of the sub-policies.
>
> * Paper [5] learns incompatible policies within a joint policy, meaning that substituting a policy in the joint policy with the incompatible policies causes a significant deterioration in performance. Our sub-policies can be viewed as a sort of incompatible policies but the measurement is a distance metric instead of performance.
>
> The core differences between our approach and those approaches are: 1. our method makes decisions with all individual policies as a whole, whereas in those methods, each individual policy makes its own decisions; 2. those works consider the diversity in the policy population as the goal, while our work focuses on the diversity of generated levels as the goal.
>
> [2] Lupu, A., Cui, B., Hu, H. & Foerster, J.. (2021). Trajectory Diversity for Zero-Shot Coordination. <i>Proceedings of the 38th International Conference on Machine Learning</i>, in <i>Proceedings of Machine Learning Research</i> 139:7204-7213 Available from https://proceedings.mlr.press/v139/lupu21a.html.
>
> [3] Cui, B., Lupu, A., Sokota, S., Hu, H., Wu, D. J., & Foerster, J. N. (2022, September). Adversarial Diversity in Hanabi. In The Eleventh International Conference on Learning Representations.
>
> [5] Charakorn, R., Manoonpong, P., & Dilokthanakul, N. (2022, September). Generating Diverse Cooperative Agents by Learning Incompatible Policies. In The Eleventh International Conference on Learning Representations.
>
> Regarding the comment **"... I think it is ok to include them as contributions to the paper, but building theoretical analysis on the interactions among ensemble policies or regularization effect would be more interesting."**: Indeed, theoretical analysis of the interactions among ensemble policies or the regularisation effect is an interesting future direction. Unfortunately, we were unable to conduct a theoretical analysis of these aspects during this rebuttal period. In our future work, we plan to delve into such theoretical analysis. For now, we provide experimental evidence showcasing the interactions among ensemble policies in Appendix E.2 of the revised paper. Specifically, we report the selection probabilities of each sub-policy during generating a level. We observe the selection probabilities of the sub-policies can be adaptively adjusted during generating levels. The selection probabilities also become more uniform as the regularisation coefficient $\lambda$ increases. We hope this added analysis can partially address your concern.

---

### Author Response · Authors · 2023-11-23
**Summary of changes**

We would like to thank all the reviewers for their careful reviews and valuable comments. We have revised the paper to address the reviewers' comments and improve the presentation and clarity. Following the reviewers' suggestions, we also added some new analysis and some examples of generated levels to Appendix. Complete generated levels of all trained generators can be found in our anonymous code repository (https://anonymous.4open.science/r/NCERL-Diverse-PCG-4F25/, the generation_results folder). The main updates of our paper are listed as below. The revisions are highlighted in blue in the paper.

1. Figure 1 and its caption have been revised as suggested by Reviewer cq3K.

2. We have removed the asynchronous framework from the contribution statements in Introduction since it is orthogonal to the main contribution of the work, as pointed out by Reviewer qX6Q.

3. We have added some additional related works to Section 2, as reviewers NGar and qX6Q have pointed out some relevant works that we did not cover in our original paper.

4. Problem formulation: Following the comments by Reviewer qX6Q and Reviewer cq3K, Section 3 has been revised to better describe our considered level generation environment, and Section 6.1 "Online Level Generation Tasks" has been revised to better explain the reward function used in our work.

5. We have clarified the description of Wasserstein distance in Section 3.2 and added Appendix C.2 to provide the formulation of Wasserstein distance as suggested by Reviewer cq3K.

6. Tables 5 and 6 have been added in Appendix E.1 to show the performance of each independent NCERL generator to better analyse the training results and answer Reviewer cq3K's question.

7. Some illustrative examples have been added to Appendix E.2 (Tables 7 and 8) to showcase the selection probabilities of sub-policies to address Reviewer qX6Q's questions.

8. We have added a comparison of levels generated by SAC and levels generated by some NCERL generators (Figures 6-11 in Appendix E.3). Complete generation results are available in our anonymous code repository.

---

### Meta-Review · Area_Chair_qQCV · 2023-12-15

**Metareview:**

This paper addresses the problem of generating a diverse sequence of levels or level segments that also correspond to certain quality metrics. This is a particular form of quality-diversity problem, dealing with generating a sequence of environments. Casting it as essentially a hierarchical RL problem, and looking for negatively correlated sub-policies, is clever. While this may seem to be a somewhat niche problem, I think if might also be useful in generating sequences of educational materials, and possibly for open-ended learning.

The paper is well-written and the results are convincing. I think it should be accepted.

**Justification For Why Not Higher Score:**

The problem is of less general relevance than some others.

**Justification For Why Not Lower Score:**

Well-done paper describing a novel approach; no real issues.

---

### Decision · Program_Chairs · 2024-01-16

Accept (poster)